# Median Clipping for Zeroth-order Non-Smooth Convex Optimization and Multi-Armed Bandit Problem with Heavy-tailed Symmetric Noise

## Abstract

In this paper, we consider non-smooth convex optimization with a zeroth-order oracle corrupted by symmetric stochastic noise. Unlike the existing high-probability results requiring the noise to have bounded $\kappa$-th moment with $\kappa \in (1, 2]$, our results allow even heavier noise with any $\kappa > 0$, e.g., the noise distribution can have unbounded expectation. Our convergence rates match the best-known ones for the case of the bounded variance, namely, to achieve function accuracy $\varepsilon$ our methods with Lipschitz oracle require $\tilde{O}(d^2\varepsilon^{-2})$ iterations for any $\kappa > 0$. We build the median gradient estimate with bounded second moment as the mini-batched median of the sampled gradient differences. We apply this technique to the stochastic multi-armed bandit problem with heavy-tailed distribution of rewards and achieve $\tilde{O}(\sqrt{dT})$ regret. We demonstrate the performance of our zeroth-order and MAB algorithms for different $\kappa$ on synthetic and real-world data. Our methods do not lose to SOTA approaches and dramatically outperform them for $\kappa \le 1$.

## 1 Introduction

During the recent few years, stochastic optimization problems with heavy-tailed noise received a lot of attention from many researchers. In particular, heavy-tailed noise is observed in various problems, such as the training of large language models [3; 44], generative adversarial networks [13; 14], finance [35], and blockchain [43]. One of the most popular techniques for handling heavy-tailed noise in theory and practice is the gradient clipping [15; 6; 31; 34] which allows deriving high-probability bounds and considerably improves convergence even in case of light tails [37].

However, most of the mentioned works focus on the gradient-based (first-order) methods. For some problems, e.g., the multi-armed bandit [10; 1; 23; 4], only losses or function values are available, and thus, zeroth-order algorithms are required. Stochastic zeroth-order optimization is being actively studied. For a detailed overview, see the recent survey [11] and the references therein. The only existing works that handle heavy-tailed noise in convex zeroth-order optimization are [19; 20] which combine clipping and gradient smoothing [12] techniques. Under noise with bounded $\kappa$-th moment for $\kappa \in (1, 2]$, the authors obtain optimal high-probability convergence for $d$-dimensional non-smooth convex problems, i.e., function accuracy $\varepsilon$ is achieved in $\tilde{O}(\sqrt{d}\varepsilon^{-1})^{\frac{\kappa}{\kappa-1}}$ oracle calls. These rates match the optimal rates for first-order optimization [15], however, they degenerate as $\kappa \to 1$, and the convergence is not guaranteed for $\kappa = 1$.

For symmetric (and close to symmetric) heavy-tailed noise distributions, the degeneration issue can be handled via median estimators [46; 34], which are frequently used in robust mean estimation and robust machine learning [27]. In the case of first-order methods, the authors of [34] achieve better complexity guarantees and show that the narrowing of the distributions' class is essential for it. However, the possibility of application of the median estimators to the case of the zeroth-order optimization and multi-armed bandit remains open. In this paper, we address this question.

### 1.1 Contributions

**Theory I.** We propose our novel theoretical zeroth-order oracle (Assumption 4) that allows us to incorporate fine-grained features of the noise probability distributions. We use it to successfully utilize symmetry of the heavy-tailed noise and dramatically improve current convergence results.

Table 1: Number of successive iterations to achieve a function's accuracy $\varepsilon$ with high probability; unconstrained optimization via Lipschitz oracle with bounded $\kappa$-th moment. Constants $b, d, M_2'$ denote the batch size, dimensionality and the Lipschitz constant of the oracle, respectively.

|  | ZO-clipped-SSTM [20] $\kappa > 1$, $b$ oracle calls per iter. | ZO-clipped-med-SSTM (this work) $\kappa > 0$, symmetric noise, $\frac{b}{\kappa}$ calls |
|---|---|---|
| Convex | $\widetilde{\mathcal{O}}\left(\max\left\{\frac{d^{\frac{1}{4}}M_2'}{\varepsilon}, \frac{1}{b}\left(\frac{\sqrt{d}M_2'}{\varepsilon}\right)^{\frac{\kappa}{\kappa-1}}\right\}\right)$ | $\widetilde{\mathcal{O}}\left(\max\left\{\frac{d^{\frac{1}{4}}M_2'}{\varepsilon}, \frac{1}{b}\left(\frac{dM_2'}{\varepsilon}\right)^2\right\}\right)$ |
| $\mu$-str. conv. | $\widetilde{\mathcal{O}}\left(\max\left\{\frac{d^{\frac{1}{4}}M_2'}{\varepsilon}, \frac{1}{b}\left(\frac{d(M_2')^2}{\mu\varepsilon}\right)^{\frac{\kappa}{2(\kappa-1)}}\right\}\right)$ | $\widetilde{\mathcal{O}}\left(\max\left\{\frac{d^{\frac{1}{4}}M_2'}{\varepsilon}, \frac{1}{b}\frac{d^2(M_2')^2}{\mu\varepsilon}\right\}\right)$ |

**Theory II.** We propose our novel ZO-clipped-med-SSTM (§3.2) for unconstrained optimization and ZO-clipped-med-SMD (§3.3) for optimization on convex compact which successfully incorporate median clipping technique. For any symmetric heavy-tailed noise with bounded $\kappa$-th moment $\kappa > 0$, our methods achieve not degenerating convergence rates with high-probability which match the optimal rates for ZO minimization under any noise with the bounded variance. In the Table 1, we provide convergence guarantees for the unconstrained case.

**Theory III.** We propose Clipped-INF-med-SMD (§4) for the stochastic multi-armed bandit (MAB) with symmetric heavy-tailed reward distribution. For MAB with $d$ arms and time interval $T$, in Theorem 3, we obtain the $\tilde{O}(\sqrt{dT})$ bound on the regret, which is optimal and matches the lower bound $\Omega(\sqrt{dT})$ for stochastic MAB with any reward distribution and bounded variance. Moreover, this bound holds not only in expectation but with controlled large deviations.

**Practice.** We demonstrate in the series of experiments (§5) on extremely noised real and synthetic data superior performance of our methods in comparison with previously known SOTA approaches.

We compare our algorithms with previous approaches and discuss its limitations in §6.

## 2 PRELIMINARIES

In this section, we introduce general notations and assumptions on optimized functions. We also recall popular gradient smoothing and clipping techniques.

**Notations.** For vector $x \in \mathbb{R}^d$ and $p \in [1, 2]$, we define $p$-norm as $\|x\|_p \stackrel{\text{def}}{=} \left(\sum_{i=1}^{d} |x_i|^p\right)^{\frac{1}{p}}$ and its dual norm as $\|x\|_q$, where $\frac{1}{p} + \frac{1}{q} = 1$. In the case $q = \infty$, we define $\|x\|_\infty = \max_{i=1,...,d} |x_i|$. We denote the Euclidean unit ball $B_2^d \stackrel{\text{def}}{=} \{x \in \mathbb{R}^d : \|x\|_2 \leq 1\}$, the Euclidean unit sphere $S_2^d \stackrel{\text{def}}{=} \{x \in \mathbb{R}^d : \|x\|_2 = 1\}$ and the probability simplex $\Delta_+^d \stackrel{\text{def}}{=} \{x \in \mathbb{R}_+^d : \sum_{i=1}^{d} x_i = 1\}$.

Median operator $\text{Median}(\{a_i\}_{i=1}^{2m+1})$ applied to the elements sequence of the odd size $2m+1, m \in \mathbb{N}$ returns $m$-th order statistics. We also use short notation for $\max$ operator, i.e. $a \vee b \stackrel{\text{def}}{=} \max(a, b)$.

**Assumption 1** (Strong convexity). *The function $f : \mathbb{R}^d \to \mathbb{R}$ is $\mu$-strongly convex, if there exists $\mu \geq 0$ such that for all $x_1, x_2 \in \mathbb{R}^d$ and $\lambda \in [0, 1]$ :*

$$f(\lambda x_1 + (1 - \lambda)x_2) \leq \lambda f(x_1) + (1 - \lambda)f(x_2) - \frac{1}{2}\mu\lambda(1 - \lambda)\|x_1 - x_2\|_2^2,$$

*If $\mu = 0$ we say that the function is just "convex".*

**Assumption 2** (Lipschitz continuity). *The function $f : \mathbb{R}^d \to \mathbb{R}$ is $M_2$-Lipschitz continuous w.r.t. the Euclidean norm, if there exists $M_2 > 0$, such that for all $x_1, x_2 \in \mathbb{R}^d$:*

$$|f(x_1) - f(x_2)| \leq M_2\|x_1 - x_2\|_2.$$

If a differentiable function has $L$-Lipschitz gradient, we call it $L$-smooth.

**Randomized smoothing.** The main scheme that allows us to develop gradient-free methods for non-smooth convex problems is randomized smoothing [9; 12; 29; 30; 40]. For the fixed smoothing

parameter $\tau > 0$, we build a smooth approximation $\hat{f}_\tau$ for a non-smooth $f : \mathbb{R}^d \to \mathbb{R}$ as:

$$\hat{f}_\tau(x) \stackrel{\text{def}}{=} \mathbb{E}_{\mathbf{u}}[f(x + \tau \mathbf{u})], \tag{1}$$

where $\mathbf{u} \sim U(B_2^d)$ is a random vector uniformly distributed on the Euclidean unit ball $B_2^d$.

If the function $f$ is $\mu$-strongly convex (As 1) and $M_2$-Lipschitz (As 2), then the smoothed function $\hat{f}_\tau$ is $\mu$-strongly convex and $\sqrt{d}M_2/\tau$-smooth. Moreover, it does not differ from the original $f$ too much, namely, (See Lemma 2 from Appendix B.1)

$$\sup_{x \in \mathbb{R}^d} |\hat{f}_\tau(x) - f(x)| \le \tau M_2. \tag{2}$$

**Clipping.** To handle heavy-tailed noise, we use a clipping technique which clips tails of gradient's distribution. For the clipping level $\lambda > 0$ and $q-$norm, where $q \in [2, +\infty]$, we define the clipping operator `clip` for arbitrary non-zero gradient vector $g \in \mathbb{R}^d$ as follows:

$$\texttt{clip}_q(g, \lambda) = \frac{g}{\|g\|_q} \min\left(\|g\|_q, \lambda\right).$$

## 3 ZEROTH-ORDER OPTIMIZATION WITH SYMMETRIC HEAVY-TAILED NOISE

In this section, we present our novel algorithms for zeroth-order optimization with independent and Lipschitz oracles. In subsection 3.1, we introduce the problem, symmetric heavy-tailed noise assumptions and median estimation with its properties. In subsection 3.2, we propose our accelerated batched ZO-clipped-med-SSTM for unconstrained problems. In subsection 3.3, we describe our ZO-clipped-med-SMD for problems on convex compacts. All proofs are located in Appendix B.

### 3.1 THEORY

We consider a non-smooth convex optimization problem on a convex set $Q \subseteq \mathbb{R}^d$:

$$\min_{x \in Q} f(x), \tag{3}$$

where $f : \mathbb{R}^d \to \mathbb{R}$ is $d$-dimensional, $\mu$-strongly convex (As 1) and $M_2$-Lipschitz (As 2) function. A point $x^*$ denotes one of the problem's solutions. In zeroth-order setup, the optimization is performed only by accessing the pairs of function evaluations rather than sub-gradients.

**Two-point oracle.** For any two points $x, y \in \mathbb{R}^d$, an oracle returns the pair of the scalar values $f(x, \xi)$ and $f(y, \xi)$, which are noised evaluation of real values $f(x)$ and $f(y)$. Moreover, noised values have the same realization of the stochastic variable $\xi$ and can be written as

$$f(x, \xi) - f(y, \xi) = f(x) - f(y) + \phi(\xi|x, y),$$

where $\phi(\xi|x, y)$ is the stochastic noise, whose distribution depends on points $x, y$.

#### 3.1.1 NOISE DISTRIBUTION.

We propose our novel assumption on distribution of $\phi(\xi|x, y)$, induced by a random variable $\xi$. It allows us to introduce symmetry and heavy-tailed noise with bounded up to $\kappa$-th moments, $\kappa > 0$.

**Assumption 3** (Symmetric noise distribution). *Symmetry. For any two points $x, y \in \mathbb{R}^d$, noise $\phi(\xi|x, y)$ has symmetric probability density $p(u|x, y)$, i.e. $p(u|x, y) = p(-u|x, y), \forall u \in \mathbb{R}$.*

*Heavy tails. We assume that there exist $\kappa > 0$, $\gamma > 0$ and scale function $B(x, y) : \mathbb{R}^d \times \mathbb{R}^d \to \mathbb{R}$, such that $\forall u \in \mathbb{R}$ holds*

$$p(u|x, y) \le \frac{\gamma^\kappa \cdot |B(x, y)|^\kappa}{|B(x, y)|^{1+\kappa} + |u|^{1+\kappa}}. \tag{4}$$

*We consider two possible oracles:*

*Independent oracle: $\phi(\xi|x, y)$ distribution doesn't depend on points $x, y$, i.e.,*

$$\gamma \cdot B(x, y) \equiv \Delta. \tag{5}$$

*Lipschitz oracle: $\phi(\xi|x, y)$ distribution becomes more concentrated around 0 as $x, y$ become closer:*

$$|\gamma \cdot B(x, y)| \le \Delta \cdot \|x - y\|_2, \tag{6}$$

*where $\Delta > 0$ is the noise Lipschitz constant.*

This assumption covers a majority of symmetric absolutely continuous distributions with bounded up to $\kappa$-th moments. For example (Remark 5), if $\xi$ has Cauchy distribution, then one can use

- Independent oracle: $f(x, \xi) = f(x) + \xi_x, f(y, \xi) = f(y) + \xi_y$ with independent $\xi_x, \xi_y$.
- Lipschitz oracle: $f(x, \boldsymbol{\xi}) = f(x) + \langle \boldsymbol{\xi}, x \rangle, f(y, \boldsymbol{\xi}) = f(y) + \langle \boldsymbol{\xi}, y \rangle$, where $\boldsymbol{\xi}$ is $d$-dimensional random vector. Oracle gives the same realization of $\boldsymbol{\xi}$ for both $x$ and $y$.

**Comparison with previous oracles.** Our Assumption 3 is quite different from the standard assumptions from [8; 20]. We make our assumption on variable $\phi(\xi|x, y)$ with fixed $x, y$. It allows us to set and use fine-grained properties of the noise distribution, e.g., symmetry or heavy tails of particular type (4). In [20], the authors fix $\xi$ and make assumption on $x, y$. Hence, they can not access the distribution of the noise and use only the fact of having bounded $\kappa$-th moment. Nevertheless, when $\kappa \in (1, 2]$, our Assumption 3 can be reduced to the standard one with the same constant, Remark 3.

We would like to highlight the fact that the common proof techniques from previous works can not be trivially generalized to apply symmetry without our novel assumption. For example, the proof of median estimator's properties Lemma 1 is based on completely different approach. We refer to Appendix A for more details and intuition behind Assumption 3.

### 3.1.2 MEDIAN ESTIMATION.

In our pipeline, instead of minimizing the non-smooth function $f$ directly, we propose to minimize the smooth approximation $\hat{f}_\tau$ with the fixed smoothing parameter $\tau$ via first-order methods. Following (2), the solution for $\hat{f}_\tau$ is also a good approximate minimizer of $f$ when $\tau$ is sufficiently small.

Following [38], the gradient of $\hat{f}_\tau$ at point $x \in \mathbb{R}^d$ can be estimated by the vector:

$$
\begin{aligned}
g(x, \mathbf{e}, \xi) &= \frac{d}{2\tau}(f(x + \tau\mathbf{e}, \xi) - f(x - \tau\mathbf{e}, \xi))\mathbf{e} \\
&= \frac{d}{2\tau}(f(x + \tau\mathbf{e}) - f(x - \tau\mathbf{e}) + \phi(\xi|x + \tau\mathbf{e}, x - \tau\mathbf{e}))\mathbf{e},
\end{aligned}
\tag{7}
$$

where $\mathbf{e} \sim U(S_2^d)$ is a random vector uniformly distributed on the Euclidean unit sphere $S_2^d$. Moreover, $\mathbf{e}, \xi$ are independent of each other conditionally on $x$. However, the noise $\phi$ might have unbounded first and second moments. To fix this, we lighten tails of $\phi$ to obtain an unbiased estimate of $\nabla \hat{f}_\tau$. For a point $x \in \mathbb{R}^d$, we apply the component-wise median operator to $2m + 1$ samples $\{g(x, \mathbf{e}, \xi^i)\}_{i=1}^{2m+1}$ with independent $\xi^i$ and the same $x$ and $\mathbf{e}$:

$$
\texttt{Med}^m(x, \mathbf{e}, \{\xi\}) \stackrel{\text{def}}{=} \texttt{Median}(\{g(x, \mathbf{e}, \xi^i)\}_{i=1}^{2m+1}).
\tag{8}
$$

The median operator can be applied to the batch of $\{\mathbf{e}^j\}_{j=1}^b$ with batch size $b$ and further averaging:

$$
\texttt{BatchMed}_b^m(x, \{\mathbf{e}\}, \{\xi\}) \stackrel{\text{def}}{=} \frac{1}{b} \sum_{j=1}^b \texttt{Med}^m(x, \mathbf{e}^j, \{\xi\}^j).
\tag{9}
$$

For a large enough number of samples, median estimations have bounded second moment.

**Lemma 1** (Median estimation's properties). *Consider $\mu$-strongly convex (As. 1) and $M_2$-Lipschitz (As. 2) function $f$ with oracle corrupted by noise under As. 3 with $\Delta$ and $\kappa > 0$. If median size $m > \frac{2}{\kappa}$ with norm $q \in [2, +\infty]$, then $\forall x \in \mathbb{R}^d$ the median estimates (8) and (9) are unbiased, i.e.,*

$$
\mathbb{E}_{\mathbf{e}, \xi}[\texttt{Med}^m(x, \mathbf{e}, \{\xi\})] = \mathbb{E}_{\mathbf{e}, \xi}[\texttt{BatchMed}_b^m(x, \{\mathbf{e}\}, \{\xi\})] = \nabla \hat{f}_\tau(x),
$$

*and have bounded second moment, i.e.,*

$$
\mathbb{E}_{\mathbf{e}, \xi}[\|\texttt{BatchMed}_b^m(x, \{\mathbf{e}\}, \{\xi\}) - \nabla \hat{f}_\tau(x)\|_2^2] \leq \frac{\sigma^2}{b},
\tag{10}
$$

$$
\mathbb{E}_{\mathbf{e}, \xi}[\|\texttt{Med}^m(x, \mathbf{e}, \{\xi\}) - \nabla \hat{f}_\tau(x)\|_q^2] \leq \sigma^2 a_q^2, \quad a_q = d^{\frac{1}{q} - \frac{1}{2}} \min\{\sqrt{32 \ln d - 8}, \sqrt{2q - 1}\}.
\tag{11}
$$

*For **independent oracle**, we have $\sigma^2 = 8dM_2^2 + 2\left(\frac{d\Delta}{\tau}\right)^2 (2m + 1)\left(\frac{4}{\kappa}\right)^{\frac{2}{\kappa}}$, and, for **Lipschitz oracle**, we have $\sigma^2 = 8dM_2^2 + (16m + 8)d^2\Delta^2\left(\frac{4}{\kappa}\right)^{\frac{2}{\kappa}}$.*

## 3.2 ZO-clipped-med-SSTM FOR UNCONSTRAINED PROBLEMS

We present our novel ZO-clipped-med-SSTM which works on the whole space $Q = \mathbb{R}^d$ with the Euclidean norm. We base it on the first-order accelerated clipped Stochastic Similar Triangles Method (clipped-SSTM) with the optimal high-probability complexity bounds from [15]. Namely, we use its zeroth-order version ZO-clipped-SSTM from [20] with the batched median estimation (9).

---

**Algorithm 1** ZO-clipped-med-SSTM

---

**Input:** Starting point $x^0 \in \mathbb{R}^d$, number of iterations $K$, median size $m$, batch size $b$, stepsize $a > 0$, smoothing parameter $\tau$, clipping levels $\{\lambda_k\}_{k=0}^{K-1}$.

1: Set $L = \sqrt{d}M_2/\tau$, $\quad A_0 = \alpha_0 = 0$, $\quad y^0 = z^0 = x^0$.
2: **for** $k = 0, \ldots, K-1$ **do**
3: $\quad$ Set $\alpha_{k+1} = {(k+2)}/{2aL}$, $\quad A_{k+1} = A_k + \alpha_{k+1}$, $\quad x^{k+1} = \frac{A_k y^k + \alpha_{k+1} z^k}{A_{k+1}}$.
4: $\quad$ Sample independently sequences $\{\mathbf{e}\} \sim U(S_2^d)$ and $\{\xi\}$ .
5: $\quad g_{med}^{k+1} = \texttt{BatchMed}_b^m(x^{k+1}, \{\mathbf{e}\}, \{\xi\})$.
6: $\quad z^{k+1} = z^k - \alpha_{k+1} \cdot \texttt{clip}_2\left(g_{med}^{k+1}, \lambda_{k+1}\right)$, $\quad y^{k+1} = \frac{A_k y^k + \alpha_{k+1} z^{k+1}}{A_{k+1}}$.
7: **end for**
**Output:** $y^K$

---

**Theorem 1** (Convergence of ZO-clipped-med-SSTM). *Consider convex (As. 1) and $M_2$-Lipschitz (As. 2) function $f$ on $\mathbb{R}^d$ with two-point oracle corrupted by noise under As. 3 with $\Delta$ and $\kappa > 0$. We set batch size $b$, median size $m = \frac{2}{\kappa} + 1$ and initial distance $R = \|x_0 - x^*\|$. To achieve function accuracy $\varepsilon$, i.e., $f(y^K) - f(x^*) \leq \varepsilon$ with probability at least $1 - \beta$ via ZO-clipped-med-SSTM with parameters $A = \ln \frac{4K}{\beta} \geq 1$, $a = \Theta(\min\{A^2, {\sigma K^2 \sqrt{A}\tau}/{\sqrt{bd}M_2 R}\})$, $\lambda_k = \Theta({R}/{(\alpha_{k+1}A)})$ and smoothing parameter $\tau = \frac{\varepsilon}{4M_2}$, the number of iterations $K$ must be*

$$\widetilde{\mathcal{O}}\left(\frac{d^{\frac{1}{4}}M_2 R}{\varepsilon} \vee \frac{(\sqrt{d}M_2 R)^2}{b \cdot \varepsilon^2}\left(1 \vee \left(\frac{4}{\kappa}\right)^{\frac{2}{\kappa}}\frac{d\Delta^2}{\varepsilon^2}\right)\right), \widetilde{\mathcal{O}}\left(\max\left\{\frac{d^{\frac{1}{4}}M_2 R}{\varepsilon}, \frac{d(M_2^2 + d\Delta^2/\kappa^{\frac{2}{\kappa}})R^2}{b \cdot \varepsilon^2}\right\}\right),$$

*for **independent** and **Lipschitz** oracle, respectively. Each iteration requires $(2m + 1) \cdot b$ oracle calls. Moreover, with probability at least $1 - \beta$ the iterates of ZO-clipped-med-SSTM remain in the ball with center $x^*$ and radius $2R$, i.e., $\{x^k\}_{k=0}^{K+1}, \{y^k\}_{k=0}^K, \{z^k\}_{k=0}^K \subseteq \{x \in \mathbb{R}^d : \|x - x^*\|_2 \leq 2R\}$.*

For Lipschitz oracle, the first term matches the optimal bound in terms of $\varepsilon$ for the deterministic non-smooth problems [5], and the second term matches the optimal bound for zeroth-order problems with the finite variance [29]. Under "optimal bound" here, we mean the optimal bound for the problems with any noise. For the symmetric noise only, we are not aware of any proved bounds. In terms of $d$, we obtain the factor $dM_2^2 + d^2\Delta^2/\kappa^{\frac{2}{\kappa}}$ instead of $(\sqrt{d}M_2 + \sqrt{d}\Delta)^{\frac{\kappa}{\kappa-1}}$ from [20].

In case of one-point oracle, while noise $\phi$ is "small", i.e.,

$$\Delta \leq \left(\frac{\kappa}{4}\right)^{\frac{1}{\kappa}} \frac{\varepsilon}{\sqrt{d}} \tag{12}$$

convergence rate is preserved. This bound on $\Delta$ is optimal in terms of $\varepsilon$, see [25; 33; 36].

For $\mu$−strongly-convex functions with Lipschitz oracle or independent oracle with small noise, we apply the restarted version of ZO-Clipped-med-SSTM. Algorithm's description, more details and results are located in Appendix C.1.

### 3.2.1 EXTENDED CLASSES OF THE OPTIMIZED FUNCTIONS

**Remark 1** (Smooth objective). *The estimates presented in Theorem 1 can be improved by introducing a new assumption, namely the assumption that the objective function $f$ is $L$-smooth with $L > 0$: $\|\nabla f(y) - \nabla f(x)\|_2 \leq L\|y - x\|_2$, $\forall x, y \in \mathbb{R}^d$. Using this assumption, we obtain the following value of the smoothing parameter $\tau = \sqrt{\varepsilon/L}$ [see 11, the end of Section*

*4.1]. Thus, assuming smoothness and convexity (As. 1) of the objective function and assuming symmetric noise (As. 3), we obtain the following estimates for the iteration complexity:*

$$\widetilde{\mathcal{O}}\left(\max\left\{\sqrt{\frac{LR^2}{\varepsilon}}, \frac{(\sqrt{d}R)^2}{b \cdot \varepsilon^2}\left(M_2^2 \vee \left(\frac{4}{\kappa}\right)^{\frac{2}{\kappa}}\frac{dL\Delta^2}{\varepsilon}\right)\right\}\right) \text{ and } \widetilde{\mathcal{O}}\left(\max\left\{\sqrt{\frac{LR^2}{\varepsilon}}, \frac{d(M_2^2 + d\Delta^2/\kappa^{\frac{2}{\kappa}})R^2}{b \cdot \varepsilon^2}\right\}\right)$$

*for independent and Lipschitz oracle, respectively. These rates match the iteration's complexity for the full gradient coordinate-wise estimation.*

**Remark 2** (Polyak–Lojasiewicz objective). *The results of Theorem 1 can be extended to the case when the objective function satisfies the Polyak–Lojasiewicz condition via restarts: let a function $f(x)$ is differentiable and there exists constant $\mu > 0$ s.t. $\forall x \in \mathbb{R}^d$ the following inequality holds $\|\nabla f(x)\|_2^2 \geq 2\mu(f(x) - f(x^*))$. Then, assuming smoothness (see Remark 1) and Polyak–Lojasiewicz condition for the objective function and assuming symmetric noise (As. 3), we obtain the following estimates for the iteration complexity: $\widetilde{\mathcal{O}}\left(\max\left\{\frac{L}{\mu}, \frac{dL}{b\mu^2\varepsilon}\left(M_2^2 \vee \left(\frac{4}{\kappa}\right)^{\frac{2}{\kappa}}\frac{dL\Delta^2}{\varepsilon}\right)\right\}\right)$*

*and $\widetilde{\mathcal{O}}\left(\max\left\{\frac{L}{\mu}, \frac{dL(M_2^2 + d\Delta^2/\kappa^{\frac{2}{\kappa}})}{b\mu^2\varepsilon}\right\}\right)$ for independent and Lipschitz oracle, respectively.*

## 3.3 ZO-clipped-med-SMD FOR CONSTRAINED PROBLEMS

We propose our novel **ZO-clipped-med-SMD** to minimize functions on a convex compact $Q \subset \mathbb{R}^d$. We use unbatched median estimation (8) in the zeroth-order algorithm **ZO-clipped-SMD** from [19], which is based on Mirror Gradient Descent.

We define 1-strongly convex w.r.t. $p$—norm and differentiable prox-function $\Psi_p$. We denote its convex (Fenchel) conjugate and its Bregman divergence, respectively, as

$$\Psi_p^*(y) = \sup_{x \in \mathbb{R}^d}\{\langle x, y\rangle - \Psi_p(x)\}, \quad V_{\Psi_p}(y, x) = \Psi_p(y) - \Psi_p(x) - \langle \nabla\Psi_p(x), y - x\rangle.$$

---

**Algorithm 2** ZO-clipped-med-SMD

**Input:** Number of iterations $K$, median size $m$, stepsize $\nu$, prox-function $\Psi_p$, smoothing parameter $\tau$, clipping level $\lambda$.
1: $x_0 = \arg\min_{x \in Q}\Psi_p(x)$.
2: **for** $k = 0, 1, \ldots, K - 1$ **do**
3:     Sample $\mathbf{e}$ from $U(S_2^d)$ and sequence $\{\xi\}$.
4:     $g_{med}^{k+1} = \mathtt{Med}^m(x^{k+1}, \mathbf{e}, \{\xi\})$.
5:     $y_{k+1} = \nabla(\Psi_p^*)(\nabla\Psi_p(x_k) - \nu \cdot \mathtt{clip}_q(g_{med}^{k+1}, \lambda)), \quad x_{k+1} = \arg\min_{x \in Q}V_{\Psi_p}(x, y_{k+1})$.
6: **end for**

**Output:** $\overline{x}_K := \frac{1}{K}\sum_{k=0}^{K}x_k$

---

**Theorem 2.** *Consider convex (As. 1) and $M_2$-Lipschitz (As. 2) function $f$ with two-point oracle corrupted by noise under As. 3 with $\kappa > 0$. To achieve function accuracy $\varepsilon$, i.e., $f(\overline{x}_K) - f(x^*) \leq \varepsilon$ with probability at least $1 - \beta$ via **ZO-clipped-med-SMD** with median size $m = \frac{2}{\kappa} + 1$, clipping level $\lambda = \sigma a_q\sqrt{K}$, stepsize $\nu = \frac{D_{\Psi_p}}{\lambda}$, diameter $D_{\Psi_p}^2 \overset{def}{=} 2\sup_{x,y \in Q}V_{\Psi_p}(x, y)$, prox-function $\Psi_p$ and $\tau = \frac{\varepsilon}{4M_2}$, the number of iterations $K$ must be*

$$\widetilde{\mathcal{O}}\left(\frac{(\sqrt{d}M_2a_qD_{\Psi_p})^2}{\varepsilon^2}\left(1 \vee \left(\frac{4}{\kappa}\right)^{\frac{2}{\kappa}}\frac{d\Delta^2}{\varepsilon^2}\right)\right), \quad \widetilde{\mathcal{O}}\left(\frac{d(M_2^2 + d\Delta^2/\kappa^{\frac{2}{\kappa}})a_q^2D_{\Psi_p}^2}{\varepsilon^2}\right) \quad (13)$$

*for **independent** and **Lipschitz** oracle, respectively. Each iteration requires $(2m + 1)$ oracle calls.*

Bounds (13) match optimal in terms of $\varepsilon$ bounds for stochastic non-smooth optimization on convex compact with the finite variance [42]. The upper bound for $\Delta$ under which the convergence rate is preserved is the same as for unconstrained optimization (12).

For $\mu$-strongly-convex functions with Lipschitz oracle or independent oracle with small noise, we apply the restarted version of **ZO-Clipped-SMD**. Algorithm and results are located in Appendix C.2.

## 4 APPLICATION TO THE MULTI-ARMED BANDIT PROBLEM WITH HEAVY TAILS

In this section, we present our novel Clipped-INF-med-SMD algorithm for multi-armed bandit (MAB) problem with heavy-tailed rewards.

**Introduction.** The stochastic MAB problem [21] can be formulated as follows: an agent at each time step $t = 1, \ldots, T$ chooses an action $A_t$ from a given action set $\mathcal{A} = (a_1, \ldots, a_n)$ and suffers stochastic loss. For each action $a_i$, there exists a probability density function for losses $\mathbf{p}(a_i)$, and an agent doesn't know them in advance. An agent can observe losses only for one action at each step, namely, the one it chooses. At each round $t$, when action $a_i$ is chosen (i.e. $A_t = a_i$), stochastic loss $\mu_{A_t} + \xi_{A_t, t}$ sampled from $\mathbf{p}(a_i)$ independently. Agent's goal is to minimize *average regret*:

$$\mathbb{E}[\mathcal{R}_T] = \sum_{t=1}^{T} [\mu_{A_t} - \mu^*], \quad \mu^* = \min_{a_i \in \mathcal{A}} \mu_i.$$

One of the main approaches for solving the MAB problem is to use reduction to the online convex optimization problem [17; 32]. Consider stochastic linear loss functions $l_t(x_t) = \langle \mu + \xi_t, x_t \rangle$, with noise $\xi_t$ and unknown fixed vector of expected losses $\mu \in \mathbb{R}^d$. The decision variable $x_t \in \triangle_+^d$ can be viewed as the player's mixed strategy (probability distribution over arms), which they use to sample arms with the aim to minimize expected regret

$$\mathbb{E}[\mathcal{R}_T(u)] = \mathbb{E}\left[ \sum_{t=1}^{T} l_t(x_t) - \min_{u \in \triangle_+^d} \left( \sum_{t=1}^{T} l_t(u) \right) \right].$$

The player observes only sampled losses for the chosen arm, i.e., the (sub)gradient $g(x) \in \partial l(x)$ is not observed in the MAB setting, and one must use an inexact oracle instead.

**Related works.** Bandits with heavy tails were introduced in [23; 4]. The heavy noise assumption usually requires the existence of $\kappa \in (1, 2]$, such that $\mathbb{E}[\|\mu + \xi_t\|^\kappa] \leq \sigma^\kappa$ (in this work, we use different Assumption 3 with $\kappa > 0$). In [4], the authors provide lower bounds on regret $\Omega\left(\sigma d^{\frac{\kappa-1}{\kappa}} T^{\frac{1}{\kappa}}\right)$ and nearly optimal algorithmic scheme called Robust UCB. Recently, a few optimal algorithms were proposed [22; 47; 18; 7] with regret bound $\tilde{O}\left(\sigma d^{\frac{\kappa-1}{\kappa}} T^{\frac{1}{\kappa}}\right)$. HTINF [18] is an INF-type algorithm with a specific pruning procedure. Algorithm 1/2-Tsallis [47] is similar to HTINF. INF-clip [7] employs a clipping mechanism instead of pruning, it clips rewards at the initial stage of the estimator construction process, prior to applying importance weighting. The main drawback of this procedure that the importance weighting procedure can artificially produce a burst in the gradient estimator. Finally, APE [22] is a perturbation-based exploration strategy that uses a p-robust mean estimator. Its algorithmic scheme is UCB-type and is very different from our algorithm.

**Our approach.** We assume that noise $\xi_t$ satisfy Assumption 3 for some $\kappa > 0$. We construct our Clipped-INF-med-SMD (Algorithm 3) based on Online Mirror Descent, but in case of symmetric noise we can improve regret upper bounds and make it $\tilde{O}(\sqrt{dT})$ which is optimal compared to the lower bound $\Omega(\sqrt{dT})$ for stochastic MAB with the bounded variance of losses. In our algorithm, we use an importance-weighted estimator:

$$\hat{g}_{t,i} = \begin{cases} \frac{g_{t,i}}{x_{t,i}} & \text{if } i = A_t \\ 0 & \text{otherwise} \end{cases},$$

where $A_t$ is the index of the chosen (at round $t$) arm. This estimator is unbiased, i.e. $\mathbb{E}_{x_t}[\hat{g}_t] = g_t$. The main drawback of this estimator is that, in the case of small $x_{t,i}$, the value of $\hat{g}_{t,i}$ can be arbitrarily large. When the noise $g_t - \mu$ has heavy tails (i.e., $\|g_t - \mu\|_\infty$ can be large with high probability), this drawback can be amplified. That is why we use robust median estimation with further clipping.

**Theorem 3.** *Consider MAB problem where the conditional probability density function for each loss satisfies Assumption 3 with $\Delta, \kappa > 0$, and $\|\mu\|_\infty \leq R$. Then, for the period $T$, the sequence $\{x_t\}_{t=1}^T$ generated by* Clipped-INF-med-SMD *with parameters* $m = \frac{2}{\kappa} + 1$, $\tau = \sqrt{d}$, $\nu = \frac{\sqrt{(2m+1)}}{\sqrt{T(36c^2 + 2R^2)}}$, $\lambda = \sqrt{T}$ *and prox-function* $\Psi_1(x) = \psi(x) \stackrel{def}{=} 2\left(1 - \sum_{i=1}^d x_i^{1/2}\right)$ *satisfies*

$$\mathbb{E}[\mathcal{R}_T(u)] \leq \sqrt{dT} \cdot (8c^2/\sqrt{d} + 4\sqrt{(2m+1)(18c^2 + R^2)}), \quad u \in \Delta_+^d, \tag{14}$$

**Algorithm 3** Clipped-INF-med-SMD

**Input:** Time period $T$, median size $m$, stepsize $\nu$, prox-function $\Psi_p$, clipping level $\lambda$.

1: $x_0 = \arg\min_{x \in \triangle_+^d} \Psi_p(x)$.

2: Set number of iterations $K = \left\lceil \frac{T-1}{2m+1} \right\rceil$.

3: **for** $k = 0, 1, \ldots, K-1$ **do**

4: Draw $A_t$ for $2m+1$ times ($t = (2m+1)\cdot k+1, \ldots, (2m+1)\cdot(k+1)$) with $P(A_t = i) = x_{k,i}$, $i = 1, \ldots, d$ and observe rewards $g_{t,A_t}$.

5: For each observation, construct estimation $\hat{g}_{t,i} = \begin{cases} \frac{g_{t,i}}{x_{k,i}} & \text{if } i = A_t \\ 0 & \text{otherwise} \end{cases}$, $i = 1, \ldots, d$.

6: $g_{med}^{k+1} = \texttt{Median}(\{\hat{g}_t\}_{t=(2m+1)\cdot k+1}^{(2m+1)\cdot(k+1)})$.

7: $y_{k+1} = \nabla(\Psi_p^*)(\nabla\Psi_p(x_k) - \nu \cdot \texttt{clip}_q\left(g_{med}^{k+1}, \lambda\right))$, $\quad x_{k+1} = \arg\min_{x \in \triangle_+^d} V_{\Psi_p}(x, y_{k+1})$.

8: **end for**

---

*where $c^2 = (32\ln d - 8) \cdot \left(8M_2^2 + 2\Delta^2(2m+1)\left(\frac{4}{\kappa}\right)^{\frac{2}{\kappa}}\right)$. Moreover, high probability bounds from Theorem 2 also hold. Proof of Theorem 3 is located in Appendix B.3.*

## 5 NUMERICAL EXPERIMENTS

In this section, we demonstrate the superior performance of ours ZO-clipped-med-SSTM and Clipped-INF-med-SMD under heavy-tailed noise on experiments on syntactical and real-world data. Additional experiments and technical details are located in Appendix D.

### 5.1 MULTI-ARMED BANDIT

We compare our Clipped-INF-med-SMD with popular SOTA algorithms tailored to handle MAB problem with heavy tails, namely, HTINF and APE. We focus on an experiment involving only two available arms ($d = 2$). Each arm $i$ generates random losses $g_{t,i} \sim \xi_t + \beta_i$. Parameters $\beta_0 = 3, \beta_1 = 3.5$ are fixed, and independent random variables $\xi_t$ have the same probability density $p_{\xi_t}(x) = \frac{1}{3 \cdot \left(1 + \left(\frac{x}{3}\right)^2\right) \cdot \pi}$.

For all methods, we evaluate the distribution of expected regret and probability of picking the best arm over 100 runs. The results are presented in Figure 1.

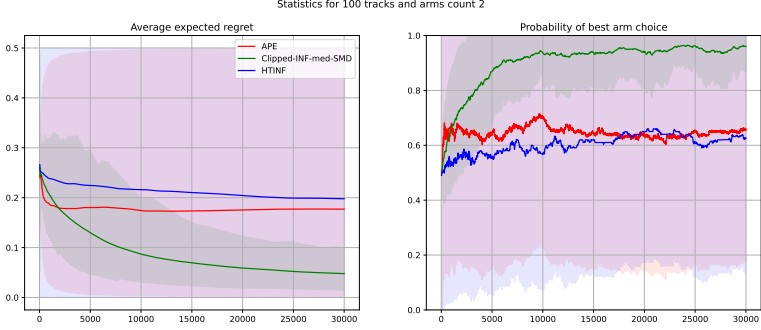

Figure 1: Average expected regret and probability of optimal arm picking mean for 100 experiments and 30000 samples with 0.95 and 0.05 percentiles for regret and ± std bounds for probabilities

As one can see from the graphs, HTINF and APE do not have convergence in probability, while our Clipped-INF-med-SMD does, which confirms the efficiency of the proposed method. In Appendix D.1, we provide technical details and additional experiments for different $\kappa$.

## 5.2 Cryptocurrency portfolio optimization

We choose cryptocurrency portfolio optimization problem for Clipped-INF-med-SMD real world application, since cryptocurrency pricing data is known by having heavy-tailed distribution. In our scenario, we have $n = 9$ assets for investing. At step $t$, we choose assets' distribution $x_{t,i} \in \Delta^n$ and then observe the whole income vector $r_{t,i}$ for each asset $i$. The main goal is to maximize total income $\max \mathbb{E} \sum_{t=1}^{T} \sum_{i=1}^{n} r_{t,i} x_{t,i}$ over a fixed time interval with length $T$.

Portfolio selection has the full feedback for all assets, while, in standard bandits, we observe only one asset per step. We adjust our Clipped-INF-med-SMD for the full feedback via calculating line 4 in Algorithm 3 for each asset $i$. As baselines, we use two strategies: hold ETH and the Efficient Frontier method [28] with maximal sharp ratio portfolio selected. For a dataset, we use open prices from Binance Spot for 2023.

The results are presented in Figure 2. As one can see, the Efficient Frontier strategy can't efficiently perform on cryptocurrency assets, and Clipped-INF-med-SMD achieved higher performance than just holding the ETH strategy, so it can be applied for detecting potentially promising assets.

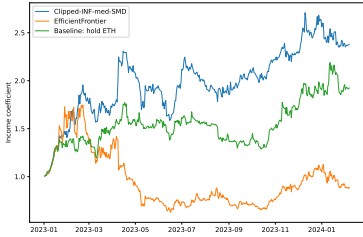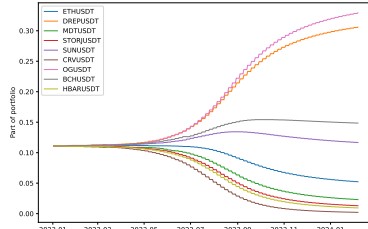

Figure 2: Strategies profit coefficient and Clipped-INF-med-SMD assets distribution over 2023 year

## 5.3 Zeroth-order optimization

To demonstrate the performance of ZO-clipped-med-SSTM, we follow [20] and conduct experiments on the following problem:

$$\min_{x \in \mathbb{R}^d} \|Ax - b\|_2 + \langle \xi, x \rangle,$$

where $\xi$ is a random vector with independent components sampled from the symmetric Levy $\alpha$-stable distribution with different $\alpha = 0.75, 1.0, 1.25, 1.5$, $A \in \mathbb{R}^{l \times d}$, $b \in \mathbb{R}^l$. Note, that $\alpha$ has the same meaning as $\kappa$, because this distribution asymptotic behavior is $f(x) \sim \frac{1}{|x|^{1+\alpha}}$ for $\alpha < 2$.

For ZO-clipped-med-SSTM, the best median size is $m = 2$. We compare it with the median size $m = 0$ which is basically ZO-clipped-SSTM. We additionally compare our algorithm with ZO-clipped-SGD from [20] and ZO-clipped-med-SGD — version of ZO-clipped-SGD with gradient estimation step replaced with median clipping version from our work.

The results over 3 launches are presented in Figure 3. The green lines on the graphs represent algorithms with median clipping. We can see that for extremely noised data $\kappa \leq 1$, our median clipping-based methods significantly outperform non-median versions. While, for standard heavy-tailed noise $\kappa > 1$, our methods do not lose to other competitors.

In Appendix D.2, we provide technical details about hyperparameters, additional experiments with enlarged number of launches and study asymmetric noise and its effect on our median methods.

**Tuning of** $m$. In experiments with both bandits and ZO methods, we grid search the median size $m$ among the range [3,5,7]. We noticed that unlike the choice of continuous the clipping level, the choice of the discrete median size only slightly affects the convergence and does not require careful fine-tuning. This range is enough to find an optimal median size for optimal convergence.

## 6 Discussion

### 6.1 Limitations

**Symmetric noise.** The assumption of the symmetric noise can be seen as a limitation from a practical point of view. It is indeed the case, but we argue that it is not as severe as it looks. A common

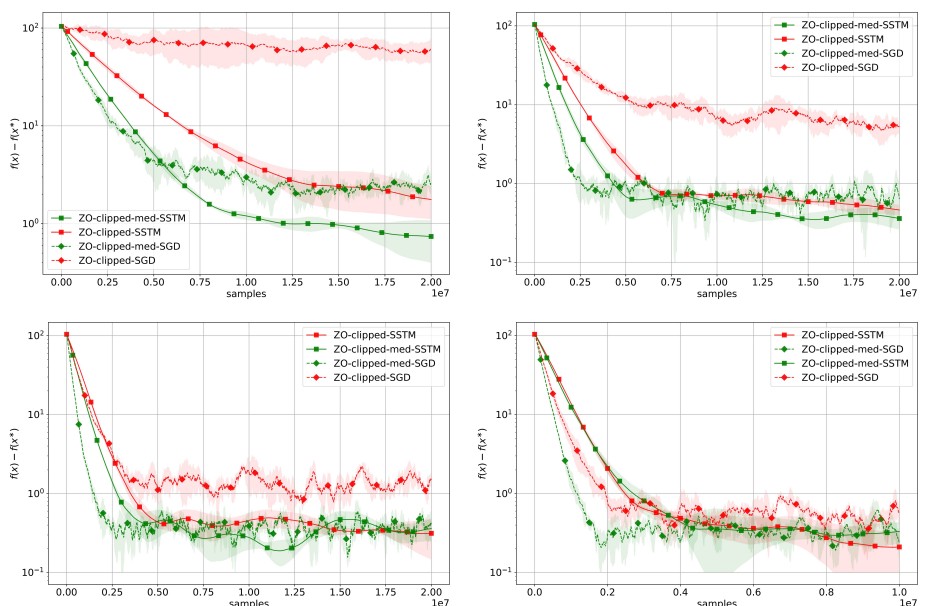

Figure 3: Convergence of ZO-clipped-SSTM, ZO-clipped-med-SSTM, ZO-clipped-SGD and ZO-clipped-med-SGD in terms of a gap function w.r.t. the number of consumed samples from the dataset for different $\alpha = \kappa$ parameters (left-to-right and top-to-bottom: 0.75, 1.0, 1.25, 1.5)

strategy to solve a general optimization problem is to run several algorithms in a competitive manner to see which performs better in practice. This approach is implemented in industrial solvers such as Gurobi. Thus, if we have different algorithms, each suited to its own conditions, we can simply test to see which one is faster for our particular case. In this scenario, we want a set of algorithms, each designed for its specific case. Our algorithm can serve as one of the options in such mix, since it provides considerable acceleration in a significant number of noise cases. Moreover, in experiments with non-symmetric noises (§D.2.1), our methods do not lose to the baselines. Hence, running our methods ends up with either typical convergence rates or faster rates for symmetric noises.

**Known $\kappa$.** In our Theorems 1, 2, 3, parameter $\kappa$ is required to set optimal median size $m = \frac{2}{\kappa} + 1$. However, for the most common cases $\kappa$ is at least 1 (i.e. expectation exists), hence we could take median size $m = 3$. In case when parameter $\kappa \to 0$, we leave the construction of an adaptive scheme [18] for future work. In practice, the choice of $m$ can be limited to a small, discrete range.

### 6.2 COMPARISON WITH PREVIOUS WORKS

Unlike the baselines ZO-clipped-SSTM [20] and APE [22], HTINF [18] with simple clipping and general heavy-tailed noise assumption $\kappa \in (1, 2]$, our Algorithms 1, 2, 3 with median clipping can work with extremely heavy-tailed noises $\kappa \le 1$. For any $\kappa > 0$, iterative complexity of our methods remains as if noise had bounded variance, namely, $\tilde{O}(d^2 \varepsilon^{-2})$ iterations to achieve function accuracy or average regret $\varepsilon$. In contrast, the best-known baselines' rates $\tilde{O}((\sqrt{d}\varepsilon^{-1})^{\frac{\kappa}{\kappa-1}})$ deteriorate depending on $\kappa$. However, such breaking results can be guaranteed only for symmetric noises, which is not as serious limitation as it seems. Nevertheless, we show that, for asymmetric noises, our methods in practice are competitive as well and perform at the same level as the baselines (§D.2.1).

### 6.3 FUTURE DIRECTIONS

**Potential impact.** We believe that ideas and obtained results from our work can inspire the community to further develop both zeroth-order methods and clipping technique. Especially considering how effectively our algorithms can work in a wide range of noise cases. For example, Lipschitz [26] and linear [39] MAB and non-convex functions [24; 41; 45] remain out of the scope of our paper.

**Broader impact**. This paper presents work which goal is to advance the field of Optimization. There are many potential societal consequences of our work, none of which we feel must be specifically highlighted here.

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

# A    REMARKS ABOUT THE ASSUMPTION ON THE NOISE

In this section, we discuss our novel noise Assumption 3. We provide comparison with previous works (Remark 3), standard examples (Remark 5) and explain the roles of parameters (Remark 4).

**Remark 3** (Comparison with previous assumptions). *In works [8; 20], different assumption on Lipschitz noise is considered. For any realization of $\xi$, the function $f(x, \xi)$ is $M_2'(\xi)$-Lipschitz, i.e.,*

$$|f(x, \xi) - f(y, \xi)| \leq M_2'(\xi)\|x - y\|_2, \quad \forall x, y \in Q \tag{15}$$

*and $M_2'(\xi)^\kappa$ has bounded $\kappa$-th moment ($\kappa > 1$), i.e., $[M_2']^\kappa \stackrel{def}{=} \mathbb{E}_\xi[M_2'(\xi)^\kappa] < \infty$.*

*We emphasize that if Assumption 3 holds with $\kappa$ then one can find $M_2'(\xi, x, y)$ such that (15) holds for any $1 < \kappa' < \kappa$ with $M_2' = O(M_2 + \Delta)$, where constant in $O(\cdot)$ depends only on $\kappa'$.*

*Proof.* Let noise $\phi(\xi|x, y)$ satisfies Assumption 3 with Lipschitz oracle and $\kappa > 1$, then it holds

$$
\begin{aligned}
|f(x, \xi) - f(y, \xi)| &= |f(x) - f(y) + \phi(\xi|x, y)| \\
&\leq |f(x) - f(y)| + |\phi(\xi|x, y)| \\
&\stackrel{\text{As } 2}{\leq} M_2\|x - y\|_2 \\
&\quad + \frac{|\phi(\xi|x, y)|}{\|x - y\|_2}\|x - y\|_2.
\end{aligned}
$$

Let us denote $M_2'(\xi, x, y) \stackrel{def}{=} M_2 + \frac{|\phi(\xi|x,y)|}{\|x-y\|_2}$ and show that for any $1 < \kappa' < \kappa$ random variable $M_2'(\xi, x, y)$ has bounded $\kappa'$-th moment which doesn't depend on $x, y$. We notice that

$$
\begin{aligned}
\mathbb{E}_\xi[|\phi(\xi|x, y)|^{\kappa'}] &= \int_{-\infty}^{+\infty} |u|^{\kappa'} p(u|x, y) du \\
&\leq \int_{-\infty}^{+\infty} \frac{|u|^{\kappa'} \gamma^\kappa |B(x, y)|^\kappa}{|B(x, y)|^{1+\kappa} + |u|^{1+\kappa}} du.
\end{aligned}
$$

After substitution $t = u/|B(x,y)|$, we get

$$
\begin{aligned}
\mathbb{E}_\xi[|\phi(\xi|x, y)|^{\kappa'}] &\leq \frac{\gamma^\kappa |B(x, y)|^\kappa}{|B(x, y)|^{\kappa-\kappa'}} \int_0^{+\infty} \frac{|t|^{\kappa'}}{1 + |t|^{1+\kappa}} dt \\
&\stackrel{(6)}{\leq} \gamma^{\kappa-\kappa'} \Delta^{\kappa'} \|x - y\|_2^{\kappa'} \int_0^{+\infty} \frac{|t|^{\kappa'}}{1 + |t|^{1+\kappa}} dt.
\end{aligned}
$$

Integral $I(\kappa') = \int_0^{+\infty} \frac{\gamma^{\kappa-\kappa'} |t|^{\kappa'} dt}{1+|t|^{1+\kappa}}$ converges since $\kappa' < \kappa$ but its value tends to $\infty$ as $\kappa' \to \kappa - 0$. Finally, we have

$$
\begin{aligned}
&\mathbb{E}_\xi[M_2'(\xi, x, y)^{\kappa'}] \\
&= \mathbb{E}_\xi\left[\left|M_2 + \frac{|\phi(\xi|x, y)|}{\|x - y\|_2}\right|^{\kappa'}\right] \\
&\stackrel{\text{Jensen inq, } \kappa' > 1}{\leq} 2^{\kappa'-1}\left[M_2^{\kappa'} + \frac{\mathbb{E}_\xi\left[|\phi(\xi|x, y)|^{\kappa'}\right]}{\|x - y\|_2^{\kappa'}}\right] \\
&\leq 2^{\kappa'-1}\left[M_2^{\kappa'} + I(\kappa')\Delta^{\kappa'}\right].
\end{aligned}
$$

Therefore, $M_2' = (\mathbb{E}_\xi[M_2'(\xi, x, y)^{\kappa'}])^{\frac{1}{\kappa'}} = O(M_2 + \Delta)$, where constant in $O(\cdot)$ depends only on $\kappa'$. $\square$

**Remark 4** (Role of the scale function $B(x,y)$). *In inequality (4) due to normalization property of probability density we must ensure that*

$$\int\limits_{-\infty}^{+\infty} \frac{\gamma^\kappa |B(x,y)|^\kappa}{|B(x,y)|^{1+\kappa} + |u|^{1+\kappa}} \, du \geq \int\limits_{-\infty}^{+\infty} p(u|x,y) du = 1.$$

*One can make substitution $t = u/|B(x,y)|$ and ensure that for $\kappa \leq 2$*

$$\int\limits_{-\infty}^{+\infty} \frac{\gamma^\kappa |B(x,y)|^\kappa du}{|B(x,y)|^{1+\kappa} + |u|^{1+\kappa}} = \gamma^\kappa \int\limits_{-\infty}^{+\infty} \frac{dt}{1 + |t|^{1+\kappa}} \overset{\kappa=1}{\geq} \gamma^\kappa \pi.$$

*Hence, $\gamma$ is sufficient to satisfy*

$$\gamma \geq \left(\frac{1}{\pi}\right)^{\frac{1}{\kappa}}.$$

*As scale value $|B(x,y)|$ decreases, quantiles of $p(u|x,y)$ gets closer to zero. Therefore, $|B(x,y)|$ can be considered as analog of variance of distribution $p(u|x,y)$.*

**Remark 5** (Standard oracles examples). *To build noise $\phi(\xi|x,y)$ satisfying Assumption 3 with $\kappa > 0$ we will use independent random variables $\{\xi_k\}$ with symmetric probability density functions $p_{\xi_k}(u)$*

$$p_{\xi_k}(u) \leq \frac{|\gamma_k \Delta_k|^\kappa}{|\Delta_k|^{1+\kappa} + |u|^{1+\kappa}}, \quad \Delta_k, \gamma_k > 0,$$

*such that for any real numbers $\{a_k\}_{k=1}^n$ and sum $\sum\limits_{k=1}^n a_k \xi_k$ it holds*

$$p_{\sum\limits_{k=1}^n a_k \xi_k}(u) \leq \frac{\left(\sum\limits_{k=1}^n |\gamma_k a_k \Delta_k|\right)^\kappa}{\left(\sum\limits_{k=1}^n |a_k \Delta_k|\right)^{1+\kappa} + |u|^{1+\kappa}}. \tag{16}$$

*Moreover, using Cauchy-Schwarz inequality we bound*

$$\sum_{k=1}^n |\gamma_k a_k \Delta_k| \leq \|(\gamma_1 \Delta_1, \ldots, \gamma_n \Delta_n)^\top\|_2 \cdot \|(a_1, \ldots, a_k)^\top\|_2. \tag{17}$$

*For example, variables $\xi_k$ can have Cauchy distribution with $\kappa = 1$ and $p(u) = \frac{1}{\pi} \frac{\Delta_k}{\Delta_k^2 + u^2}$ parametrized by scale $\Delta_k$. For the independent Cauchy variables with scales $\{\Delta_k\}_{k=1}^n$ and any real numbers $\{a_k\}_{k=1}^n$, the sum $\sum\limits_{k=1}^n a_k \xi_k$ is the Cauchy variable with scale $\sum\limits_{k=1}^n |a_k| \Delta_k$. Therefore, inequality (16) for Cauchy variables holds true. For oracles, we have the following constants.*

- *Independent oracle:*

  *$f(x,\xi) = f(x) + \xi_x, f(y,\xi) = f(y) + \xi_y, \phi(\xi|x,y) = \xi_x - \xi_y$, where $\xi_x, \xi_y$ are independent samples for each point $x$ and $y$. Thus, we have the final scale $\Delta = \Delta_x + \Delta_y$.*

- *Lipschitz oracle:*

  *$f(x,\boldsymbol{\xi}) = f(x) + \langle \boldsymbol{\xi}, x \rangle, f(y,\boldsymbol{\xi}) = f(y) + \langle \boldsymbol{\xi}, y \rangle, \phi(\boldsymbol{\xi}|x,y) = \langle \boldsymbol{\xi}, x - y \rangle$, where $\boldsymbol{\xi}$ is $d$-dimensional random vector with components $\xi_k$. Oracle gives the same realization of $\boldsymbol{\xi}$ for both $x$ and $y$. In that case, the vector $\boldsymbol{\xi}$ can be restated to $\boldsymbol{\xi} = A\boldsymbol{\xi}_{ind}$ with $\phi(\boldsymbol{\xi}|x,y) = \langle \boldsymbol{\xi}_{ind}, A^\top (x-y) \rangle$, where $A$ is the correlation matrix and $\boldsymbol{\xi}_{ind}$ are independent Cauchy variables. Now, if the vector $\boldsymbol{\xi}_{ind}$ has scales $\{\Delta_k\}_{k=1}^n$, then we have $\gamma$ and $B(x,y)$ from Assumption 3 equal to*

$$\gamma = \frac{1}{\pi},$$

$$B(x,y) = \sum_{k=1}^d |\Delta_k [A^\top (x-y)]_k| \overset{(17)}{\leq} \|(\Delta_1, \ldots, \Delta_d)^\top\|_2 \|A^\top\|_2 \|x-y\|_2.$$

## B  PROOFS

### B.1  PROOF OF LEMMA 1.

To begin with, we need some properties of the smoothed approximation $\hat{f}_\tau$.

**Lemma 2** ([12], Theorem 2.1). *Consider $\mu$-strongly convex (As. 1) and $M_2$-Lipschitz (As. 2) function $f$. For the smoothed function $\hat{f}_\tau$ defined in (1), the following properties hold true:*

    *1. Function $\hat{f}_\tau$ is $M_2$-Lipschitz and satisfies*

$$\sup_{x \in \mathbb{R}^d} |\hat{f}_\tau(x) - f(x)| \leq \tau M_2.$$

    *2. Function $\hat{f}_\tau$ is differentiable on $\mathbb{R}^d$ with the following gradient at point $x \in \mathbb{R}^d$:*

$$\nabla \hat{f}_\tau(x) = \mathbb{E}_{\mathbf{e}}\left[\frac{d}{\tau} f(x + \tau\mathbf{e})\mathbf{e}\right],$$

    *where $\mathbf{e} \sim U(S_2^d)$ is a random vector uniformly distributed on the unit Euclidean sphere.*

    *3. Function $\hat{f}_\tau$ is $L$-smooth with $L = \sqrt{d}M_2/\tau$ on $\mathbb{R}^d$.*

**Proposition 1** (Strong convexity of $\hat{f}_\tau$). *Consider $\mu$-strongly convex (As. 1) function $f$. Then the smoothed function $\hat{f}_\tau$ defined in (1) is also $\mu$-strongly convex.*

*Proof.* Function $f$ is $\mu$-strongly convex if for any points $x, y \in \mathbb{R}^d$ and $t \in [0, 1]$ we have

$$f(xt + y(1 - t)) \leq t \cdot f(x) + (1 - t) \cdot f(y) - \frac{1}{2}\mu t(1 - t)\|x - y\|_2^2.$$

Following definition of $\hat{f}_\tau$, we write down for $\mathbf{u} \in U(B_2^d)$ inequality

$$
\begin{aligned}
f(xt + y(1-t) + \tau\mathbf{u}) &= f((x + \tau\mathbf{u}) \cdot t + (y + \tau\mathbf{u}) \cdot (1 - t)) \\
&\leq t \cdot f(x + \tau\mathbf{u}) + (1 - t) \cdot f(y + \tau\mathbf{u}) - \frac{1}{2}\mu t(1 - t)\|x - y\|_2^2.
\end{aligned}
$$

Taking math expectation $\mathbb{E}_{\mathbf{u}}$ from both sides, we have

$$\mathbb{E}_{\mathbf{u}}[f(xt + y(1 - t) + \tau\mathbf{u})] \leq t \cdot \mathbb{E}_{\mathbf{u}}[f(x + \tau\mathbf{u})] + (1 - t) \cdot \mathbb{E}_{\mathbf{u}}[f(y + \tau\mathbf{u})] - \frac{1}{2}\mu t(1 - t)\|x - y\|_2^2.$$

$\square$

*Proof of Lemma 1.* Firstly, we notice from our construction of the oracle

$$f(x, \xi) - f(y, \xi) = f(x) - f(y) + \phi(\xi|x, y), \quad \forall x, y \in \mathbb{R}^d,$$

we have

$$
\begin{aligned}
g(x, \mathbf{e}, \xi) &= \frac{d}{2\tau}(f(x + \tau\mathbf{e}, \xi) - f(x - \tau\mathbf{e}, \xi)) \\
&= \frac{d}{2\tau}[f(x + \tau\mathbf{e}) - f(x - \tau\mathbf{e})]\mathbf{e} + \frac{d}{2\tau}\phi(\xi|x + \tau\mathbf{e}, x - \tau\mathbf{e})\mathbf{e}
\end{aligned}
$$

and for $\texttt{Med}^m(x, \mathbf{e}, \{\xi\})$ we have

$$
\begin{aligned}
\texttt{Med}^m(x, \mathbf{e}, \{\xi\}) &= \texttt{Median}\left(\left\{g(x, \mathbf{e}, \xi^i)\right\}_{i=1}^{2m+1}\right) \\
&= \texttt{Median}\left(\left\{\frac{d}{2\tau}[f(x + \tau\mathbf{e}) - f(x - \tau\mathbf{e})]\mathbf{e} + \frac{d}{2\tau}\phi(\xi^i|x + \tau\mathbf{e}, x - \tau\mathbf{e})\mathbf{e}\right\}_{i=1}^{2m+1}\right) \\
&= \frac{d}{2\tau}[f(x + \tau\mathbf{e}) - f(x - \tau\mathbf{e})]\mathbf{e} \qquad\qquad (18) \\
&+ \frac{d}{2\tau}\texttt{Median}\left(\left\{\phi(\xi^i|x + \tau\mathbf{e}, x - \tau\mathbf{e})\right\}_{i=1}^{2m+1}\right)\mathbf{e}. \qquad (19)
\end{aligned}
$$

**Finite second moment:**

Further, we analyze two terms: gradient estimation term (18) and the noise term (19).

Following work [19] [Lemma 2.3.] we have an upper bound for the second moment of (18)

$$\mathbb{E}_{\mathbf{e}}\left[\left\|\left\|\frac{d}{2\tau}[f(x+\tau\mathbf{e})-f(x-\tau\mathbf{e})]\mathbf{e}\right\|\right\|_q^2\right] \leq da_q^2 M_2^2, \tag{20}$$

where $a_q = d^{\frac{1}{q}-\frac{1}{2}}\min\{\sqrt{32\ln d-8}, \sqrt{2q-1}\}$ is a special coefficient, such that,

$$\mathbb{E}_{\mathbf{e}}[\|\mathbf{e}\|_q^2] \leq a_q^2. \tag{21}$$

See Lemma 2.1 from [16] and Lemma 8.4 from [19] for more details.

Next, we deal with noise term (19). For symmetric variable $\phi(\xi|x,y)$ for all $x,y \in \mathbb{R}^d$ under Assumption 3 it holds

$$p(u) \leq \frac{\gamma^\kappa|B(x,y)|^\kappa}{|B(x,y)|^{1+\kappa}+|u|^{1+\kappa}}.$$

Further, we prove that, for large enough $m$, noise term has finite variance. For this purpose, we denote $Y \stackrel{\text{def}}{=} \texttt{Median}\left(\{\phi(\xi^i|x,y)\}_{i=1}^{2m+1}\right)$ and cumulative distribution function of $Y$

$$P(t) \stackrel{\text{def}}{=} \int_{-\infty}^{t} p(u)du.$$

Median of $2m+1$ i.i.d. variables distributed according to $p(u)$ is $(m+1)$-th order statistic, which has probability density function

$$(2m+1)\binom{2m}{m}P(t)^m(1-P(t))^m p(t).$$

The second moment $\mathbb{E}[Y^2]$ can be calculated via

$$\begin{aligned}
\mathbb{E}[Y^2] &= \int_{-\infty}^{+\infty}(2m+1)\binom{2m}{m}t^2 P(t)^m(1-P(t))^m p(t)dt \\
&\leq (2m+1)\binom{2m}{m}\sup_t\{t^2 P(t)^m(1-P(t))^m\}\int_{-\infty}^{+\infty}p(t)dt \\
&\leq (2m+1)\binom{2m}{m}\sup_t\{t^2 P(t)^m(1-P(t))^m\}.
\end{aligned}$$

For any $t < 0$, we have

$$\begin{aligned}
P(t) &= \int_{-\infty}^{t}p(u)du \leq \int_{-\infty}^{t}\frac{|\gamma B(x,y)|^\kappa}{|B(x,y)|^{1+\kappa}+|u|^{1+\kappa}} \\
&\leq \int_{-\infty}^{t}\frac{|\gamma B(x,y)|^\kappa}{|u|^{1+\kappa}} \leq \frac{|\gamma B(x,y)|^\kappa}{\kappa}\cdot\frac{1}{|t|^\kappa}.
\end{aligned}$$

Similarly, one can prove that for any $t > 0$

$$1-P(t) = \int_{t}^{\infty}p(u)du \leq \frac{|\gamma B(x,y)|^\kappa}{\kappa}\cdot\frac{1}{t^\kappa}.$$

Since for any number $a \in [0, 1]$ holds $a(1 - a) \le \frac{1}{4}$ we have for any $t \in \mathbb{R}$

$$P(t)(1 - P(t)) \le \min\left\{\frac{1}{4}, \frac{|\gamma B(x, y)|^\kappa}{\kappa} \cdot \frac{1}{|t|^\kappa}\right\}$$

along with

$$t^2 P(t)^m (1 - P(t))^m \le \min\left\{\frac{t^2}{4^m}, \left(\frac{|\gamma B(x, y)|^\kappa}{\kappa}\right)^m \cdot \frac{1}{|t|^{m\kappa - 2}}\right\}. \tag{22}$$

If $m\kappa > 2$ the first term of (22) increasing and the second one decreasing with the growth of $|t|$, then the maximum of the minimum (22) is achieved when

$$\frac{t^2}{4^m} = \left(\frac{|\gamma B(x, y)|^\kappa}{\kappa}\right)^m \cdot \frac{1}{|t|^{m\kappa - 2}},$$

$$|t| = |\gamma B(x, y)| \left(\frac{4}{\kappa}\right)^{\frac{1}{\kappa}}.$$

Therefore, we get for any $t \in \mathbb{R}$

$$t^2 P(t)^m (1 - P(t))^m \le \frac{|\gamma B(x, y)|^2}{4^m} \left(\frac{4}{\kappa}\right)^{\frac{2}{\kappa}},$$

and, as a consequence

$$\mathbb{E}[Y^2] \le (2m + 1)\binom{2m}{m}\frac{|\gamma B(x, y)|^2}{4^m}\left(\frac{4}{\kappa}\right)^{\frac{2}{\kappa}}.$$

It only remains to note

$$\binom{2m}{m} = \frac{(2m)!}{m! \cdot m!} = \prod_{j=1}^{m}\frac{2j}{j} \cdot \prod_{j=1}^{m}\frac{2j - 1}{j} \le 4^m.$$

Since $Y$ has the finite second moment, it has finite math expectation

$$\mathbb{E}[Y] = \int_{-\infty}^{+\infty} (2m + 1)\binom{2m}{m} t P(t)^m (1 - P(t))^m p(t)dt.$$

For any $t \in \mathbb{R}$, due to symmetry of $p(t)$, we have $P(t) = (1 - P(-t))$ and $p(t) = p(-t)$ and, as a consequence,

$$\mathbb{E}[Y] = \int_{-\infty}^{+\infty} (2m + 1)\binom{2m}{m} t P(t)^m (1 - P(t))^m p(t)dt = 0.$$

Finally, we have an upper bound for (19)

$$\mathbb{E}_{\mathbf{e}, \xi}\left\|\frac{d}{2\tau}\mathtt{Median}\big(\{\phi(\xi^i | x + \tau\mathbf{e}, x - \tau\mathbf{e})\}\big)\mathbf{e}\right\|_q^2$$

$$= \left(\frac{d}{2\tau}\right)^2 \mathbb{E}_{\mathbf{e}}[\mathbb{E}_\xi[Y^2 | \mathbf{e}] \cdot \|\mathbf{e}\|_q^2]$$

$$\le \left(\frac{d}{2\tau}\right)^2 (2m + 1)\left(\frac{4}{\kappa}\right)^{\frac{2}{\kappa}} \cdot \mathbb{E}_{\mathbf{e}}[|\gamma B(x + \tau\mathbf{e}, x - \tau\mathbf{e})|^2 \|\mathbf{e}\|_q^2]. \tag{23}$$

In case of the **independent** oracle, from Assumption 3 and (5) we simplify

$$\mathbb{E}_{\mathbf{e}}[|\gamma B(x + \tau\mathbf{e}, x - \tau\mathbf{e})|\|\mathbf{e}\|_q^2] \leq \Delta^2 \mathbb{E}_{\mathbf{e}}[\|\mathbf{e}\|_q^2] \overset{(21)}{\leq} \Delta^2 a_q^2. \tag{24}$$

In case of the **Lipschitz** oracle, we use (6) and get

$$\mathbb{E}_{\mathbf{e}}[|\gamma B(x + \tau\mathbf{e}, x - \tau\mathbf{e})|\|\mathbf{e}\|_q^2] \quad \leq \quad 4\Delta^2\tau^2\mathbb{E}_{\mathbf{e}}[\|\mathbf{e}\|_2^2\|\mathbf{e}\|_q^2] \overset{(21)}{\leq} 4\Delta^2\tau^2 a_q^2.$$

Combining upper bounds (20) and (24) or (25), we obtain total bound

$$\mathbb{E}_{\mathbf{e},\xi}[\|\texttt{Med}^m(x, \mathbf{e}, \{\xi\})\|_q^2] \quad \leq \quad 2 \cdot (20) + 2 \cdot (24)$$
$$(25).$$

For the batched gradient estimation $\texttt{BatchMed}_b^m(x, \{\mathbf{e}\}, \{\xi\})$ and $q = 2$, we use Lemma 4 from [20] that states

$$\mathbb{E}_{\mathbf{e},\xi}[\|\texttt{BatchMed}_b^m(x, \{\mathbf{e}\}, \{\xi\})\|_2^2] \leq \frac{1}{b} \cdot \mathbb{E}_{\mathbf{e},\xi}[\|\texttt{Med}^m(x, \mathbf{e}, \{\xi\})\|_2^2].$$

For the bound of the centered second moment, we use Jensen's inequality for any random vector $X$
$$\mathbb{E}[\|X - \mathbb{E}[X]\|_q^2] \leq 2\mathbb{E}[\|X\|_q^2] + 2\|\mathbb{E}[X]\|_q^2 \leq 4\mathbb{E}[\|X\|_q^2].$$

**Unbiasedness:**

According to Lemma 2, the term (18) is an unbiased estimation of the gradient $\nabla \hat{f}_\tau(x)$. Indeed, the distribution of $\mathbf{e}$ is symmetrical and we can derive

$$\mathbb{E}_{\mathbf{e}}\left[\frac{d}{2\tau}[f(x + \tau\mathbf{e}) - f(x - \tau\mathbf{e})]\mathbf{e}\right] = \mathbb{E}_{\mathbf{e}}\left[\frac{d}{\tau}[f(x + \tau\mathbf{e})]\right] = \nabla \hat{f}^\tau(x).$$

Since $Y$ has the finite second moment, it has finite math expectation

$$\mathbb{E}[Y] = \int_{-\infty}^{+\infty} (2m + 1)\binom{2m}{m} tP(t)^m(1 - P(t))^m p(t)dt.$$

For any $t \in \mathbb{R}$, due to symmetry of $p(t)$, we have $P(t) = (1 - P(-t))$ and $p(t) = p(-t)$ and, as a consequence,

$$\mathbb{E}[Y] = \int_{-\infty}^{+\infty} (2m + 1)\binom{2m}{m} tP(t)^m(1 - P(t))^m p(t)dt = 0.$$

Hence, we obtained that $\mathbb{E}_{\mathbf{e},\xi}[\texttt{Med}^m(x, \mathbf{e}, \{\xi\})] = \nabla \hat{f}_\tau(x)$ along with $\mathbb{E}_{\mathbf{e},\xi}[\texttt{BatchMed}_b^m(x, \{\mathbf{e}\}, \{\xi\})] = \nabla \hat{f}_\tau(x)$ as the batching is the mean of random vectors with the same math expectation. □

## B.2 PROOF OF CONVERGENCE THEOREMS 1 AND 2

For any point $x \in \mathbb{R}^d$, we might consider median estimations $\texttt{Med}^m(x, \mathbf{e}, \{\xi\})$ and $\texttt{BatchMed}_b^m(x, \{\mathbf{e}\}, \{\xi\})$ to be the oracle for the gradient of $\hat{f}_\tau(x)$ that satisfies Assumption 4.

**Assumption 4.** *Let $G(x, \mathbf{e}, \xi)$ be the oracle for the gradient of function $\hat{f}_\tau(x)$, such that for any point $x \in Q$ it is unbiased, i.e.,*

$$\mathbb{E}_{\mathbf{e},\xi}[G(x, \mathbf{e}, \xi)] = \nabla \hat{f}_\tau(x),$$

*and has bounded second moment, i.e.,*

$$\mathbb{E}_{\mathbf{e},\xi}[\|G(x, \mathbf{e}, \xi) - \nabla \hat{f}_\tau(x)\|_q^2] \leq \Sigma_q^2, \tag{25}$$

*where $\Sigma_q$ might depend on $\tau$.*

Thus, in order to prove convergence of ZO-clipped-med-SSTM and ZO-clipped-med-SMD we use general convergence theorems with oracle satisfying Assumption 4 for ZO-clipped-SSTM (Theorem 1 from [20] with $\alpha = 2$) and ZO-clipped-SMD (Theorem 4.3 from [19] with $\kappa = 1$). Next, we take $\texttt{BatchMed}_b^m(x, \{\mathbf{e}\}, \{\xi\})$ and $\texttt{Med}^m(x, \mathbf{e}, \{\xi\})$ as the necessary oracles and substitute $\Sigma_q$ from (25) with $\sigma$ and $\sigma a_q$ from Lemma 1, respectively.

### B.2.1 UNCONSTRAINED PROBLEMS.

**Theorem 4** (Convergence of ZO-clipped-SSTM). *Consider convex (As. 1) and $M_2$-Lipschitz (As. 2) function $f$ on $\mathbb{R}^d$ with gradient oracle under As. 4 with $\Sigma_2$.*

*We denote $\|x^0 - x^*\|_2^2 \le R^2$, where $x^0$ is a starting point and $x^*$ is an optimal solution to (3).*

*We run ZO-clipped-SSTM for $K$ iterations with smoothing parameter $\tau$, batch size $b$, probability $1 - \beta$ and further parameters $A = \ln \frac{4K}{\beta} \ge 1$, $a = \Theta(\min\{A^2, \Sigma_2 K^2 \sqrt{A}\tau/\sqrt{db}M_2 R\})$, $\lambda_k = \Theta(R/(\alpha_{k+1}A))$. We guarantee that with probability at least $1 - \beta$:*

$$f(y^k) - f(x^*) = 2M_2\tau + \widetilde{\mathcal{O}}\left(\max\left\{\frac{\sqrt{d}M_2 R^2}{\tau K^2}, \frac{\Sigma_2 R}{\sqrt{bK}}\right\}\right).$$

*Moreover, with probability at least $1 - \beta$ the iterates of ZO-clipped-SSTM remain in the ball with center $x^*$ and radius $2R$, i.e., $\{x^k\}_{k=0}^{K+1}, \{y^k\}_{k=0}^K, \{z^k\}_{k=0}^K \subseteq \{x \in \mathbb{R}^d : \|x - x^*\|_2 \le 2R\}$.*

For ZO-clipped-med-SSTM, optimal convergence rate and parameters are presented in Theorem 5.

**Theorem 5** (Convergence of ZO-clipped-med-SSTM). *Consider convex (As. 1) and $M_2$-Lipschitz (As. 2) function $f$ on $\mathbb{R}^d$ with two-point oracle corrupted by noise under As. 3 with $\kappa > 0$.*

*We denote $\|x^0 - x^*\|_2^2 \le R^2$, where $x^0$ is a starting point and $x^*$ is an optimal solution to (3).*

*We run ZO-clipped-med-SSTM for $K$ iterations with smoothing parameter $\tau$, batchsize $b$, probability $1 - \beta$ and further parameters $m = \frac{2}{\kappa} + 1, A = \ln \frac{4K}{\beta} \ge 1$, $a = \Theta(\min\{A^2, \sigma K^2 \sqrt{A}\tau/\sqrt{bd}M_2 R\})$, $\lambda_k = \Theta(R/(\alpha_{k+1}A))$. We guarantee that with probability at least $1 - \beta$:*

$$f(y^k) - f(x^*) = 2M_2\tau + \widetilde{\mathcal{O}}\left(\max\left\{\frac{\sqrt{d}M_2 R^2}{\tau K^2}, \frac{\sigma R}{\sqrt{bK}}\right\}\right), \tag{26}$$

*where $\sigma$ comes from Lemma 1.*

*Moreover, with probability at least $1 - \beta$ the iterates of ZO-clipped-med-SSTM remain in the ball with center $x^*$ and radius $2R$, i.e., $\{x^k\}_{k=0}^{K+1}, \{y^k\}_{k=0}^K, \{z^k\}_{k=0}^K \subseteq \{x \in \mathbb{R}^d : \|x - x^*\|_2 \le 2R\}$.*

Statement of Theorem 1 follows if we equate both terms of (26) to $\frac{\varepsilon}{2}$, taking $\tau = \frac{\varepsilon}{4M_2}$ and explicit formula for $\sigma$ from Lemma 1.

### B.2.2 CONSTRAINED PROBLEMS.

**Theorem 6** (Convergence of ZO-clipped-SMD). *Consider convex (As. 1) and $M_2$-Lipschitz (As. 2) function $f$ on a convex compact $Q$ with gradient oracle under As. 4 with $\Sigma_q$.*

*We run ZO-clipped-SMD for $K$ iterations with smoothing parameter $\tau$, norm $q \in [2, +\infty]$, prox-function $\Psi_p$, probability $1 - \beta$ and further parameters $\lambda = \Sigma_q\sqrt{K}$, $\nu = \frac{D_{\Psi_p}}{\lambda}$, where squared diameter $D_{\Psi_p}^2 \stackrel{def}{=} 2 \sup_{x,y \in Q} V_{\Psi_p}(x, y)$. We guarantee that with probability at least $1 - \beta$:*

$$f(y^k) - f(x^*) = 2M_2\tau + \widetilde{\mathcal{O}}\left(\frac{\Sigma_q D_{\Psi_p}}{\sqrt{K}}\right).$$

For ZO-clipped-med-SMD, optimal convergence rate and parameters are presented in Theorem 7.

**Theorem 7** (Convergence of ZO-clipped-med-SMD). *Consider convex (As. 1) and $M_2$-Lipschitz (As. 2) function $f$ on a convex compact $Q$ with two-point oracle corrupted by noise As. 3 with $\kappa > 0$.*

*We run* ZO-clipped-med-SMD *for $K$ iterations with smoothing parameter $\tau$, $q \in [2, +\infty]$, prox-function $\Psi_p$, probability $1 - \beta$ and further parameters $m = \frac{2}{\kappa} + 1$, $\lambda = \sigma a_q \sqrt{K}$, $\nu = \frac{D_{\Psi_p}}{\lambda}$, where diameter squared $D_{\Psi_p}^2 \overset{def}{=} 2 \sup\limits_{x, y \in Q} V_{\Psi_p}(x, y)$. We guarantee that with probability at least $1 - \beta$:*

$$f(y^k) - f(x^*) = 2M_2 \tau + \widetilde{\mathcal{O}}\left(\frac{\sigma a_q D_{\Psi_p}}{\sqrt{K}}\right),$$

*where $\sigma, a_q$ come from Lemma 1.*

Statement of Theorem 2 follows if we equate both terms of (27) to $\frac{\varepsilon}{2}$, taking $\tau = \frac{\varepsilon}{4M_2}$ and explicit formulas for $\sigma$ and $a_q$ from Lemma 1.

**Recommendations for standard constrained problems.** In this paragraph, we discuss some standard sets $Q$ and prox-functions $\Psi_p$ taken from [2]. We can choose prox-functions to reduce $a_q D_{\Psi_p}$ and get better convergence constants. The two main setups are

1. Ball setup, $p = 2, q = 2$:
$$\Psi_p(x) = \frac{1}{2}\|x\|_2^2,$$

2. Entropy setup, $p = 1, q = \infty$:

$$\Psi_p(x) = (1 + \gamma) \sum_{i=1}^{d} (x_i + \gamma/d) \log(x_i + \gamma/d).$$

We consider unit balls $B_{p'}^d$ and standard simplex $\triangle_+^d$ as $Q$. For $Q = \triangle_+^d$ or $B_1^d$, the Entropy setup is preferable. Meanwhile, for $Q = B_2^d$ or $B_\infty^d$, the Ball setup is better.

### B.3 PROOF OF THEOREM 3

**Lemma 3.** *Let $f(x)$ be a linear function, then $\nabla f(x) = \nabla \hat{f}_\tau(x)$.*

*Proof.*
$$\nabla \hat{f}_\tau(x) = \nabla \mathbb{E}_{\mathbf{u} \sim B_2^d}[f(x + \tau \mathbf{u})] = \nabla \mathbb{E}_{\mathbf{u} \sim B_2^d}[\langle \mu, x + \tau \mathbf{u} \rangle]$$
$$= \nabla \langle \mu, x + \tau \mathbb{E}_{\mathbf{u} \sim B_2^d}[u] \rangle = \nabla \langle \mu, x \rangle = \nabla f(x).$$

$\square$

**Lemma 4.** *Let $f(x)$ be a linear function, $q = \infty$, $\tau = \alpha \sqrt{d}$, then*

$$\mathbb{E}_{\mathbf{e}, \xi}[\|g_{med}^{k+1} - \mu\|_\infty^2] \leq (32 \ln d - 8) \cdot \left(8M_2^2 + 2\alpha^2 \Delta^2 (2m + 1)\left(\frac{4}{\kappa}\right)^{\frac{2}{\kappa}}\right).$$

*Proof.* From 1 with $q = \infty$ and $\tau = \alpha \sqrt{d}$ we get

$$\mathbb{E}_{\mathbf{e}, \xi}[\|\mathrm{Med}^m(x, \mathbf{e}, \{\xi\}) - \nabla \hat{f}_\tau(x)\|_\infty^2] \leq \sigma^2 a_\infty^2, \quad a_\infty = d^{-\frac{1}{2}}\sqrt{32 \ln d - 8},$$

where $\sigma^2 = d\left(8M_2^2 + 2\alpha^2 \Delta^2 (2m + 1)\left(\frac{4}{\kappa}\right)^{\frac{2}{\kappa}}\right)$.

Hence, w.r.t (3) we get

$$\mathbb{E}_{\mathbf{e}, \xi}[\|g_{med}^{k+1} - \mu\|_\infty^2] \leq (32 \ln d - 8) \cdot \left(8M_2^2 + 2\alpha^2 \Delta^2 (2m + 1)\left(\frac{4}{\kappa}\right)^{\frac{2}{\kappa}}\right).$$

$\square$

**Lemma 5.** *[Lemma 5.1 from [37]]* *Let $X$ be a random vector in $\mathbb{R}^d$ and $\bar{X} = \mathtt{clip}(X, \lambda)$, then*

$$\|\bar{X} - \mathbb{E}[\bar{X}]\| \leq 2\lambda. \tag{27}$$

*Moreover, if for some $c \geq 0$*

$$\mathbb{E}[X] = x \in \mathbb{R}^n, \quad \mathbb{E}[\|X - x\|^2] \leq c^2$$

*and $\|x\| \leq \frac{\lambda}{2}$, then*

$$\left\|\mathbb{E}[\bar{X}] - x\right\| \leq \frac{4c^2}{\lambda}, \tag{28}$$

$$\mathbb{E}\left[\left\|\bar{X} - x\right\|^2\right] \leq 18c^2, \tag{29}$$

$$\mathbb{E}\left[\left\|\bar{X} - \mathbb{E}[\bar{X}]\right\|^2\right] \leq 18c^2. \tag{30}$$

**Remark 6.** *Combination of Lemma 4 and Lemma 5 with $X = g_{med}^{k(t)}$ and $x = \mu$ in case when $\lambda \geq 2\|\mu\|_\infty$ immidiatly get the following bounds:*

$$\left\|\mathbb{E}[g_{med}^{k(t)}] - \mathbb{E}[\tilde{g}_{med}^{k(t)}]\right\|_\infty = \left\|\mu - \mathbb{E}[\tilde{g}_{med}^{k(t)}]\right\|_\infty \leq \frac{4c^2}{\lambda},$$

$$\mathbb{E}\left[\|\tilde{g}_{med}^{k(t)}\|_\infty^2\right] \leq 2\mathbb{E}\left[\|\tilde{g}_{med}^{k(t)} - \mu\|_\infty^2 + \|\mu\|_\infty^2\right] \leq 2\|\mu\|_\infty^2 + 36c^2,$$

*for $c^2 = (32\ln d - 8) \cdot \left(8M_2^2 + 2\alpha^2\Delta^2(2m+1)\left(\frac{4}{\kappa}\right)^{\frac{2}{\kappa}}\right)$.*

**Lemma 6.** *Suppose that* Clipped-INF-med-SMD *with $1/2$-Tsallis entropy*

$$\psi(x) = 2\left(1 - \sum_{i=1}^d x_i^{1/2}\right), \quad x \in \Delta_+^d$$

*as prox-function generates the sequences $\{x_k\}_{k=0}^K$ and $\{\tilde{g}_{med}^k\}_{k=0}^K$, then for any $u \in \Delta_+^d$ holds:*

$$\sum_{k=0}^K \sum_{s=1}^{2m+1} \langle \tilde{g}_{med}^k, x_k - u\rangle$$

$$\leq (2m+1)\left[2\frac{d^{1/2} - \sum_{i=1}^d u_i^{1/2}}{\nu} + \nu\sum_{k=0}^K\sum_{i=1}^d (\tilde{g}_{med}^k)_i^2 \cdot x_{k,i}^{3/2}\right].$$

*Proof.* By definition, the *Bregman divergence* $V_\psi(x, y)$ is:

$$V_\psi(x, y) = \psi(x) - \psi(y) - \langle \nabla\psi(y), x - y\rangle$$

$$= 2\left(1 - \sum_{i=1}^d x_i^{1/2}\right) - 2\left(1 - \sum_{i=1}^d y_i^{1/2}\right) + \sum_{i=1}^d y_i^{-1/2}(x_i - y_i)$$

$$= -2\sum_{i=1}^d x_i^{1/2} + 2\sum_{i=1}^d y_i^{1/2} + \sum_{i=1}^d y_i^{-1/2}(x_i - y_i).$$

Note that the algorithm can be considered as an online mirror descent (OMD) with batching and the Tsallis entropy used as prox-function:

$$x_{k+1} = \arg\min_{x \in \Delta_+^d}\left[\nu x^{\mathsf{T}}\tilde{g}_{med}^k + V_\psi(x, x_k)\right].$$

Thus, the standard inequality for OMD holds:

$$\langle \tilde{g}_{med}^k, x_k - u\rangle \leq \frac{1}{\nu}\left[V_\psi(u, x_k) - V_\psi(u, x_{k+1}) - V_\psi(x_{k+1}, x_k)\right] + \langle \tilde{g}_{med}^k, x_k - x_{k+1}\rangle. \tag{31}$$

From Tailor theorem, we have

$$V_\psi(z, x_k) = \frac{1}{2}(z - x_k)^T \nabla^2 \psi(y_k)(z - x_k) = \frac{1}{2}\|z - x_k\|^2_{\nabla^2 \psi(y_k)}$$

for some point $y_k \in [z, x_k]$.

Hence, we have

$$\langle \tilde{g}^k_{med}, x_k - x_{k+1}\rangle - \frac{1}{\nu}V_\psi(x_{k+1}, x_k)$$

$$\leq \max_{z \in R^d_+}\left[\langle \tilde{g}^k_{med}, x_k - z\rangle - \frac{1}{\nu}V_\psi(z, x_k)\right]$$

$$= \left[\langle \tilde{g}^k_{med}, x_k - z^*_k\rangle - \frac{1}{\nu}V_\psi(z^*_k, x_k)\right]$$

$$\leq \frac{\nu}{2}\|\tilde{g}^k_{med}\|^2_{(\nabla^2 \psi(y_k))^{-1}} + \frac{1}{2}\|z^* - x_k\|^2_{\nabla^2 \psi(y_k)} - \frac{1}{\nu}V_\psi(z^*, x_k)$$

$$= \frac{\nu}{2}\|\tilde{g}^k_{med}\|^2_{(\nabla^2 \psi(y_k))^{-1}},$$

where $z^* = \arg\max_{z \in \mathbb{R}^d_+}\left[\langle \tilde{g}^k_{med}, x_k - z\rangle - \frac{1}{\nu}V_\psi(z, x_k)\right]$.

Proceeding with (31), we get:

$$\langle \tilde{g}^k_{med}, x_k - u\rangle \leq \frac{1}{\nu}\left[V_\psi(u, x_k) - V_\psi(u, x_{k+1})\right] + \frac{\nu}{2}\|\tilde{g}^k_{med}\|^2_{(\nabla^2 \psi(y_k))^{-1}}.$$

Sum over $k$ gives

$$\sum_{k=0}^{K}\langle \tilde{g}^k_{med}, x_k - u\rangle$$

$$\leq \frac{V_\psi(x_0, u)}{\nu} + \frac{\nu}{2}\sum_{k=0}^{K}(\tilde{g}^k_{med})^T\left(\nabla^2 \psi(y_k)\right)^{-1}\tilde{g}^k_{med}$$

$$= 2\frac{d^{1/2} - \sum_{i=1}^{d}u_i^{1/2}}{\nu} + \nu\sum_{k=0}^{K}\sum_{i=1}^{d}(\tilde{g}^k_{med})^2_i y^{3/2}_{k,i}, \tag{32}$$

where $y_k \in [x_k, z^*_k]$ and $z^*_k = \arg\max_{z \in R^d_+}\left[\langle \tilde{g}^k_{med}, x_k - z\rangle - \frac{1}{\nu}V_\psi(z, x_k)\right]$.

From the first-order optimality condition for $z^*_k$ we obtain

$$-\nu(\tilde{g}^k_{med})_i + (x_{k,i})^{1/2} = (z^*_{k,i})^{1/2}$$

and thus we get $z^*_{k,i} \leq x_{k,i}$.

Thus, (32) becomes

$$\sum_{k=0}^{K}\langle \tilde{g}^k_{med}, x_k - u\rangle \leq 2\frac{d^{1/2} - \sum_{i=1}^{d}u_i^{1/2}}{\nu} + \nu\sum_{k=0}^{K}\sum_{i=1}^{d}(\tilde{g}^k_{med})^2_i \cdot x^{3/2}_{k,i}$$

and concludes the proof. $\qquad\square$

**Lemma 7.** *Suppose that* Clipped-INF-med-SMD *with $1/2$-Tsallis entropy as prox-function generates the sequences $\{x_k\}_{k=0}^{K}$ and $\{\tilde{g}^k_{med}\}_{k=0}^{K}$, and for each arm $i$ random reward $g_{t,i}$ at any step $t$ has bounded expectation $\mathbb{E}[g_{t,i}] \leq \frac{\lambda}{2}$ and the noise $g_{t,i} - \mu_i$ has symmetric distribution, then for any $u \in \Delta^d_+$ holds:*

$$\mathbb{E}_{x_k, \mathbf{e}_{[k]}, \xi_{[k]}}\left[\sum_{i=1}^{d}(\tilde{g}^k_{med})^2_i \cdot x^{3/2}_{k,i}\right] \leq \sqrt{d} \cdot (2\|\mu\|^2_\infty + 36c^2). \tag{33}$$

*Proof:*

$$\mathbb{E}_{x_k,\mathbf{e}_{[k]},\xi_{[k]}}\left[\sum_{i=1}^{d}(\tilde{g}_{med}^{k})_i^2 \cdot x_{k,i}^{3/2}\right] \le \mathbb{E}_{x_k,\mathbf{e}_{[k]},\xi_{[k]}}\left[\sum_{i=1}^{d}(\tilde{g}_{med}^{k})_i^2 \cdot x_{k,i}^{1/2}\right]$$

$$\le \mathbb{E}_{x_k,\mathbf{e}_{[k]},\xi_{[k]}}\left[\sqrt{\sum_{i=1}^{d}(\tilde{g}_{med}^{k})_i^2}\cdot\sqrt{(\tilde{g}_{med}^{k})_i^2\cdot x_{k,i}^{1/2}}\right]$$

$$\le \sqrt{\mathbb{E}_{x_k,\mathbf{e}_{[k]},\xi_{[k]}}\left[\sum_{i=1}^{d}(\tilde{g}_{med}^{k})_i^2\right]}\cdot\sqrt{\mathbb{E}_{x_k,\mathbf{e}_{[k]},\xi_{[k]}}\left[(\tilde{g}_{med}^{k})_i^2\cdot x_{k,i}^{1/2}\right]}$$

$$\le \sqrt{d}\cdot(2\|\mu\|_\infty^2 + 36c^2).$$

**Theorem 3** Consider MAB problem where the conditional probability density function for each loss satisfies Assumption 3 with $\Delta, \kappa > 0$, and $\|\mu\|_\infty \le R$. Then, for the period $T$, the sequence $\{x_t\}_{t=1}^T$ generated by Clipped-INF-med-SMD with parameters $m = \frac{2}{\kappa} + 1$, $\tau = \alpha\sqrt{d}$, $\nu = \frac{\sqrt{(2m+1)}}{\sqrt{T(36c^2+2R^2)}}$, $\lambda = \sqrt{T}$ and prox-function $\psi(x) = 2\left(1 - \sum_{i=1}^d x_i^{1/2}\right)$ satisfies

$$\mathbb{E}\left[\mathcal{R}_T(u)\right] \le \sqrt{Td}\cdot(8c^2/\sqrt{d} + 4\sqrt{(2m+1)(18c^2 + R^2)}),\quad u \in \Delta_+^d, \tag{34}$$

where $c^2 = (32\ln d - 8)\cdot\left(8M_2^2 + 2\alpha^2\Delta^2(2m+1)\left(\frac{4}{\kappa}\right)^{\frac{2}{\kappa}}\right)$. Moreover, high probability bounds from Theorem 2 also hold.

*Proof of Theorem 3:* Firstly, for any $x, y \in \triangle_+^d$ we have

$$\|x - y\|_2 \le \sqrt{2}. \tag{35}$$

Next we obtain

$$\mathbb{E}\left[\mathcal{R}_T(u)\right] = \mathbb{E}\left[\sum_{t=1}^{T}l(x_t) - \sum_{t=1}^{T}l(u)\right] \le \mathbb{E}\left[\sum_{t=1}^{T}\langle\nabla l(x_t), x_t - u\rangle\right]$$

$$\le \mathbb{E}\left[\sum_{t=1}^{T}\langle\mu - g_{med}^{k(t)}, x_{k(t)} - u\rangle\right] + \mathbb{E}\left[\sum_{t=1}^{T}\langle g_{med}^{k(t)} - \tilde{g}_{med}^{k(t)}, x_{k(t)} - u\rangle\right] + \mathbb{E}\left[\sum_{t=1}^{T}\langle\tilde{g}_{med}^{k(t)}, x_{k(t)} - u\rangle\right]$$

$$= \mathbb{E}\left[\sum_{t=1}^{T}\langle g_{med}^{k(t)} - \tilde{g}_{med}^{k(t)}, x_{k(t)} - u\rangle\right] + \mathbb{E}\left[\sum_{t=1}^{T}\langle\tilde{g}_{med}^{k(t)}, x_{k(t)} - u\rangle\right]$$

$$\le \left[\sum_{t=1}^{T}\|\mathbb{E}[g_{med}^{k(t)}] - \mathbb{E}[\tilde{g}_{med}^{k(t)}]\|_\infty \cdot \|x_{k(t)} - u\|_1\right] + \mathbb{E}\left[\sum_{t=1}^{T}\langle\tilde{g}_{med}^{k(t)}, x_{k(t)} - u\rangle\right]$$

$$\overset{\text{Remark 6, (35)}}{\le} \frac{8c^2 T}{\lambda} + (2m+1)\mathbb{E}\left[\sum_{k=0}^{K}\langle\tilde{g}_{med}^{k}, x_k - u\rangle\right]$$

$$\overset{\text{Lemma 6}}{\le} \frac{8c^2 T}{\lambda} + (2m+1)\left[2\frac{d^{1/2} - \sum_{i=1}^d u_i^{1/2}}{\nu} + \nu\sum_{k=0}^{K}\sum_{i=1}^{d}((\tilde{g}_{med}^{k})_i^2 \cdot x_{k,i}^{3/2}\right]$$

$$\overset{\text{Lemma 7}}{\le} \frac{8c^2 T}{\lambda} + 2(2m+1)\frac{\sqrt{d}}{\nu} + \nu T\sqrt{d}(36c^2 + 2\|\mu\|_\infty^2)$$

$$= \sqrt{Td}\cdot(8c^2/\sqrt{d} + 4\sqrt{(2m+1)(18c^2 + R^2)}),$$

where $c^2 = (32\ln d - 8)\cdot\left(8M_2^2 + 2\alpha^2\Delta^2(2m+1)\left(\frac{4}{\kappa}\right)^{\frac{2}{\kappa}}\right)$.

$\square$

# C   RESTARTED ALGORITHMS ZO-clipped-SSTM AND ZO-clipped-SMD.

The restart technique is to run in cycle algorithm $\mathcal{A}$, taking the output point from the previous run as the initial point for the current one.

---

**Algorithm 4** Restarted ZO-clipped-$\mathcal{A}$

---

**Input:** Starting point $x^0$, number of restarts $N_r$, number of iterations $\{K_t\}_{t=1}^{N_r}$, algorithm $\mathcal{A}$, parameters $\{P_t\}_{t=1}^{N_r}$.
1: $\hat{x}^0 = x^0$.
2: **for** $t = 1, \ldots, N_r$ **do**
3:     Run algorithms $\mathcal{A}$ with parameters $P_t$ and starting point $\hat{x}^{t-1}$. Set output point as $\hat{x}^t$.
4: **end for**
**Output:** $\hat{x}^{N_r}$

---

Strong convexity of function $f$ with minimum $x^*$ implies an upper bound for the distance between point $x$ and solution $x^*$ as

$$\frac{\mu}{2}\|x - x^*\|_2^2 \le f(x) - f(x^*).$$

Considering upper bounds from Corollary 1, 2 for $f(x) - f(x^*)$, one can construct a relation between $\|x_0 - x^*\|_2$ and $\|x - x^*\|_2$ after $K$ iterations. Based on this relation, one can calculate iteration, after which it is more efficient to start a new run rather than continue current with slow convergence rate.

We apply the general Convergence Theorem 2 from [20] for R-ZO-clipped-SSTM and Theorem 5.2 from [19] for R-ZO-clipped-SMD with oracle satisfying Assumption 4. However, oracle can not depend on, $\tau$ which means that we should use either Lipschitz oracle or one-point oracle with small noise, i.e.,

$$\Delta \le \left(\frac{\kappa}{4}\right)^{\frac{1}{\kappa}} \frac{\varepsilon}{\sqrt{d}}. \tag{36}$$

In the Convergence Theorems, minimal necessary value of $\tau = \frac{\varepsilon}{4M_2}$, hence

$$\sigma^2 = 8dM_2^2 + 2\left(\frac{d\Delta}{\tau}\right)^2 (2m+1)\left(\frac{4}{\kappa}\right)^{\frac{2}{\kappa}} \le 32(2m+1) \cdot dM_2^2.$$

## C.1   UNCONSTRAINED PROBLEMS

**Theorem 8** (Convergence of R-ZO-clipped-SSTM). *Consider $\mu$-strongly convex (As. 1) and $M_2$-Lipschitz (As. 2) function $f$ on $\mathbb{R}^d$ with gradient oracle under As. 4 with $\Sigma_2$.*

*We denote $\|x^0 - x^*\|^2 \le R^2$, where $x^0$ is a starting point.*

*Let $\varepsilon$ be desired accuracy, value $1 - \beta$ be desired probability and $N_r = \lceil \log_2(\mu R^2/2\varepsilon) \rceil$ be the number of restarts. For each stage $t = 1, ..., N_r$, we run ZO-clipped-SSTM with batch size $b_t$, $\tau_t = \varepsilon_t/4M_2$, $L_t = M_2\sqrt{d}/\tau_t$, $K_t = \widetilde{\Theta}(\max\{\sqrt{L_t R_{t-1}^2/\varepsilon_t}, (\Sigma_2 R_{t-1}/\varepsilon_t)^2/b_t\})$, $a_t = \widetilde{\Theta}(\max\{1, \Sigma_2 K_t^{\frac{3}{2}}/\sqrt{b_t}L_t R_t\})$ and $\lambda_k^t = \widetilde{\Theta}(R/\alpha_{k+1}^t)$, where $R_{t-1} = 2^{-\frac{(t-1)}{2}}R$, $\varepsilon_t = \mu R_{t-1}^2/4$, $\ln 4N_r K_t/\beta \ge 1$, $\beta \in (0,1]$. Then, to guarantee $f(\hat{x}^{N_r}) - f(x^*) \le \varepsilon$ with probability at least $1 - \beta$, R-ZO-clipped-SSTM requires*

$$\widetilde{\mathcal{O}}\left(\max\left\{\sqrt{\frac{M_2^2\sqrt{d}}{\mu\varepsilon}}, \frac{\Sigma_2^2}{\mu\varepsilon}\right\}\right) \tag{37}$$

*total number of oracle calls.*

**Theorem 9** (Convergence of Restarted ZO-clipped-med-SSTM). *Consider $\mu$-strongly convex (As. 1) and $M_2$-Lipschitz (As. 2) function $f$ on $\mathbb{R}^d$ with oracle corrupted by noise under As. 3 with $\Delta, \kappa > 0$.*

*To achieve function accuracy $\varepsilon$, i.e., $f(\hat{x}^{N_r}) - f(x^*) \le \varepsilon$ with probability at least $1 - \beta$ via Restarted ZO-clipped-med-SSTM median size must be $m = \frac{2}{\kappa} + 1$, other parameters must be set according to Theorem 8 ($\Sigma_2 = \sigma$ from Lemma 1). Then, Restarted ZO-clipped-med-SSTM requires for*

- *independent oracle under* (36)*:*

$$\widetilde{\mathcal{O}} \left( (2m+1) \cdot \max \left\{ \sqrt{\frac{M_2^2 \sqrt{d}}{\mu \varepsilon}}, \frac{d M_2^2}{\kappa \mu \varepsilon} \right\} \right), \tag{38}$$

- *Lipschitz oracle:*

$$\widetilde{\mathcal{O}} \left( (2m+1) \cdot \max \left\{ \sqrt{\frac{M_2^2 \sqrt{d}}{\mu \varepsilon}}, \frac{d(M_2^2 + d\Delta^2/\kappa^{\frac{2}{\kappa}})}{\mu \varepsilon} \right\} \right) \tag{39}$$

*total number of oracle calls.*

Similar to the convex case, the first term in bounds (39), (38) matches the optimal in $\varepsilon$ bound for the deterministic case for non-smooth strongly convex problems (see [5]). The second term matches the optimal in terms of $\varepsilon$ bound for zeroth-order problems with finite variance (see [29]).

## C.2   CONSTRAINED PROBLEMS

**Theorem 10** (Convergence of R-ZO-clipped-SMD)**.** *Consider $\mu$-strongly convex (As. 1) and $M_2$-Lipschitz (As. 2) function $f$ on a convex compact $Q$ with gradient oracle under As. 4 with $\Sigma_q$.*

*We set the prox-function $\Psi_p$ and norm $p \in [1, 2]$. Denote $R_0^2 \stackrel{def}{=} \sup_{x,y \in Q} 2 V_{\Psi_p}(x, y)$ for the diameter of the set $Q$ and $R_t = R_0/2^t$.*

*Let $\varepsilon$ be desired accuracy and $N = \widetilde{O} \left( \frac{1}{2} \log_2 \left( \frac{\mu R_0^2}{2\varepsilon} \right) \right)$ be the number of restarts. For each $t = \overline{1, N_r}$, we run ZO-clipped-SMD with $K_t = \widetilde{O} \left( \left[ \frac{\Sigma_q}{\mu R_t} \right]^2 \right)$, $\tau_t = \frac{\Sigma_q R_t}{M_2 \sqrt{K_t}}$, $\lambda_t = \sqrt{K_t} \Sigma_q$ and $\nu_t = \frac{R_t}{\lambda_t}$. To guarantee $f(\hat{x}^{N_r}) - f(x^*) \le \varepsilon$ with prob. at least $1 - \beta$, R-ZO-clipped-SMD requires*

$$\widetilde{O} \left( \frac{\Sigma_q^2}{\mu \varepsilon} \right)$$

*total number of oracle calls.*

**Theorem 11** (Convergence of Restarted ZO-clipped-med-SMD)**.** *Consider $\mu$-strongly convex (As. 1) and $M_2$-Lipschitz (As. 2) function $f$ on $\mathbb{R}^d$ with two-point oracle corrupted by noise under As. 3 with $\kappa > 0$ and $\Delta > 0$.*

*To achieve accuracy $\varepsilon$, i.e., $f(\hat{x}^{N_r}) - f(x^*) \le \varepsilon$ via Restarted ZO-clipped-med-SMD with probability at least $1 - \beta$ median size must be $m = \frac{2}{\kappa} + 1$, other parameters must be set according to Theorem 10 ($\Sigma_q = \sigma a_q$ from Lemma 1). In this case, Restarted ZO-clipped-med-SMD requires for*

- *independent oracle under* (36)*:*

$$\widetilde{\mathcal{O}} \left( (2m+1) \cdot \frac{d M_2^2 a_q^2}{\kappa \mu \varepsilon} \right), \tag{40}$$

- *Lipschitz oracle:*

$$\widetilde{\mathcal{O}} \left( (2m+1) \cdot \frac{d(M_2^2 + d\Delta^2/\kappa^{\frac{2}{\kappa}}) a_q^2}{\mu \varepsilon} \right) \tag{41}$$

*total number of oracle calls, where $a_q = d^{\frac{1}{q} - \frac{1}{2}} \min\{\sqrt{32 \ln d - 8}, \sqrt{2q - 1}\}$.*

## D EXPERIMENTS DETAILS

Each experiment is computed on a CPU in several hours. The code is written in Python and will be made public after acceptance. For HTINF [18], APE [22], ZO-clipped-SSTM and ZO-clipped-SGD [20], we provide our own implementation based on pseudocodes from the original articles.

### D.1 MULTI-ARMED BANDITS

In our experimental setup, individual experiments are subject to significant random deviations. To enhance the informativeness of the results, we conduct 100 individual experiments and analyze aggregated statistics.

By design, we possess knowledge of the conditional probability of selecting the optimal arm for all algorithms, which remains stochastic due to the nature of the experiment's history.

To mitigate the high dispersion in probabilities, we apply an average filter with a window size of 30 to reduce noise in the plot. APE and HTINF can't handle cases when noise expectation is unbounded, so we modeled this case with a low value of $\alpha = 0.01$, where $1 + \alpha$ is the moment that exists in the problem statement for APE and HTINF.

#### D.1.1 DEPENDENCE ON $\kappa$

We conduct experiments to check dependence on $\kappa$ under the symmetric Levy $\alpha$-stable noise, where $\alpha = \kappa$. We compare standard INFC method from [7] which allows $\kappa \leq 1$ with Clipped-INF-med-SMD, and comparison results can be found in Figure 4.

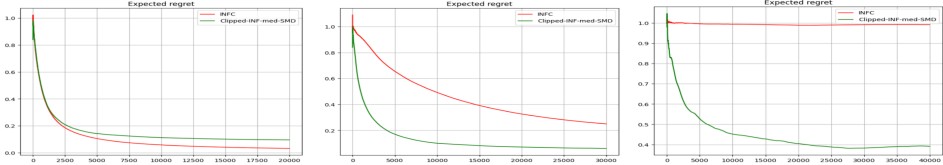

Figure 4: Convergence of Clipped-INF-med-SMD and INFC under $\kappa = 1.5, 1, 0.5$, respectively

### D.2 ZEROTH-ORDER OPTIMIZATION

To generate $A \in \mathbb{R}^{l \times d}$ and $b \in \mathbb{R}^l$ we draw them from standard normal distribution with $d = 16$ and $l = 200$. For algorithms, we gridsearch stepsize $a$ over $\{0.1, 0.01, 0.001, 0.0001\}$ and smoothing parameter $\tau$ over $\{0.1, 0.01, 0.001\}$. For ZO-clipped-med-SSTM, the parameters $a = 0.001$, $L = 1$ (note that $a$ and $L$ are actually used together in the algorithm, therefore, we gridsearch only one of them) and $\tau = 0.01$ are the best. For ZO-clipped-med-SGD, we use $a = 0.01$, default momentum of 0.9 and $\tau = 0.1$. For non-median versions, after the same gridsearch, parameters happened to be the same.

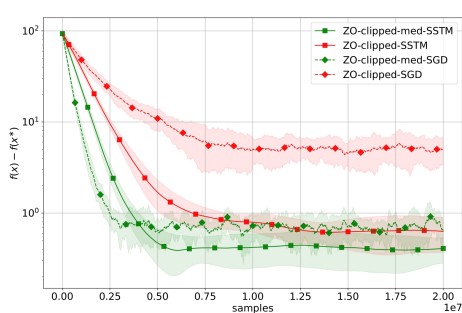

Figure 5: Convergence of ZO-clipped-SSTM, ZO-clipped-med-SSTM, ZO-clipped-SGD and ZO-clipped-med-SGD over 15 launches

To obtain better estimates for methods' performance, we conduct experiment with $\kappa = 1$ over 15 launches and present the results in Figure 5.

#### D.2.1 SYMMETRIC AND ASYMMETRIC NOISE

To check the dependence on the addition of an asymmetric part to the noise, we replace the noise $\xi$ with $\xi = w * \xi_1 + (1 - w) * |\xi_2|$ with $\xi_1$ drawn from a symmetric Levy $\alpha$-stable distribution with $\alpha = 1.0$ and $\xi_2$ being a random vector with independent components sampled from

- the same distribution
- standard normal distribution.

For $w$, we consider $0.9$ (meaning the weight of symmetric noise is bigger) and $0.5$ (equal impact). We take a component-wise absolute values of $\xi_2$, which makes $w$ a mix of symmetric and asymmetric noise. The results are presented in Figures 6 (Levy noise) and 7 (normal noise).

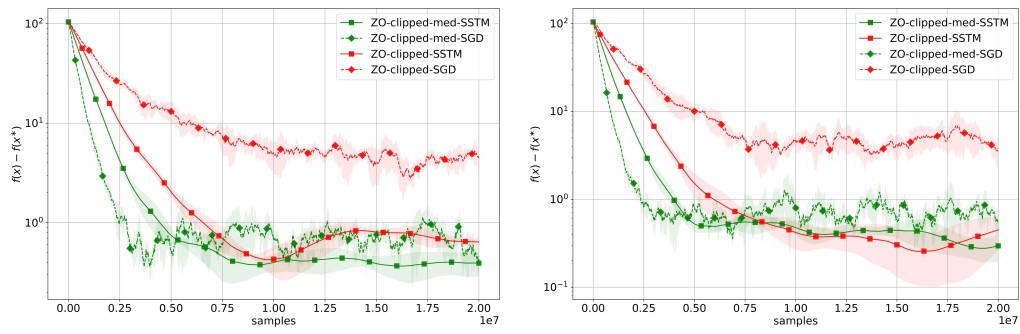

Figure 6: Convergence of ZO-clipped-SSTM, ZO-clipped-med-SSTM, ZO-clipped-SGD and ZO-clipped-med-SGD with asymmetric Levy noise addition with weight of symmetric part of $0.9$ and $0.5$ on left and right, respectively

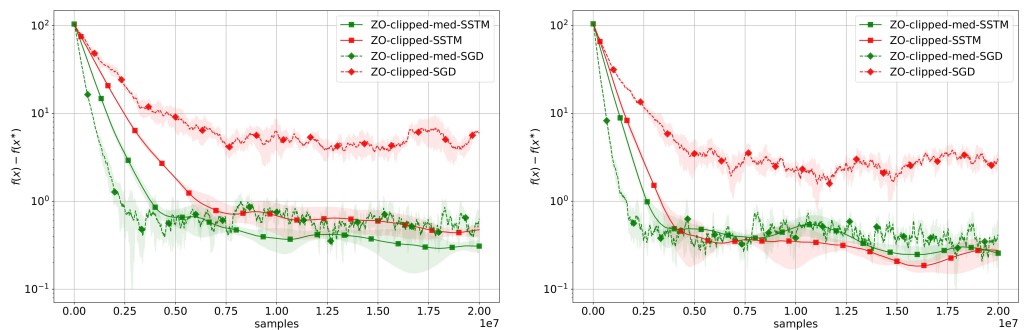

Figure 7: Convergence of ZO-clipped-SSTM, ZO-clipped-med-SSTM, ZO-clipped-SGD and ZO-clipped-med-SGD with asymmetric normal noise addition with weight of symmetric part of $0.9$ and $0.5$ on left and right, respectively

