# OpenReview forum: "Median Clipping for Zeroth-order Non-Smooth Convex Optimization and Multi Arm Bandit Problem with Heavy-tailed Symmetric Noise"
_ICLR.cc/2025/Conference — Submitted to ICLR 2025_

### Official Review · Reviewer_Jooe · 2024-10-29

**Soundness:** 3
**Presentation:** 3
**Contribution:** 2
**Rating:** 5
**Confidence:** 3

**Summary:**

This paper considers zeroth-order convex optimization problems with symmetric stochastic heavy-tailed noise. The gradient is estimated by the median in a batch, instead of using the empirical mean used in prior works. This technique is also applied to the linear bandits with heavy-tailed symmetric noise.

**Strengths:**

- With the additional symmetric noise assumption, this paper has improved the prior methods under the heavy-tailed noise case.
- The paper is relatively clear and easy to follow. Most concepts are well explained and the comparison of the final results with prior works is clear.

**Weaknesses:**

**Majors**:
- Under the additional symmetric noise assumption, this paper deals with the case when $\kappa\in(0,1)$, and improves the bound for $\kappa\in[1,2)$. From the algorithm design, Algorithm 1 and ZO-clipped-SSTM (Kornilov et al. (2023b)) largely follow the same procedure, expect that the estimate of the gradient changes from the randomly smoothed mean to the randomly smoothed median. Similarly, Algorithm 2 and ZO-Clip-SMD (Kornilov et al. (2023a)) are similar expect for the gradient estimate part. From this perspective, the main procedure required to deal with $\kappa\in(0,1)$ is the change of estimation statistics. It would be great if the authors can highlight the additional technical challenges and novelties. Otherwise, this paper looks incremental given the prior works.
- The motivation for the symmetric assumption is not well explained. As the symmetry in the noise is critical for the median estimate, a motivation is needed to support this assumption.
- As this paper adopts bandits as an application and there is already some work in the bandits literature which considers heavy-tailed noise or robust bandits problems (e.g., [1]) without the symmetric noise assumption, it would be great if the authors can compare their results to them, in terms of the assumptions, methods, theoretical guarantees and efficiency of the algorithms.

**Minors**:
- Can the authors kindly modify Line 73? It can be misleading as Bubeck et al. (2013) does not make the symmetric noise assumption.

[1] Mathieu, T., Basu, D., & Maillard, O. A. (2024). Bandits Corrupted by Nature: Lower Bounds on Regret and Robust Optimistic Algorithms. _Transactions on Machine Learning Research Journal_.

**Questions:**

- For the stochastic MAB problem, as suggested by the authors that the regret upper bound of the proposed algorithm is $\tilde{O}(\sqrt{Td})$, which is suboptimal compared to the lower bound for MAB with bounded variance of losses. Is this because the proposed algorithm can deal with unbounded variance case (with the symmetric noise assumption)? If it is, how is the proposed upper bound in Theorem 3? Is there any lower bound?
- According to Theorem 1, 2, 3, the parameter $m$, which decides the batch size used to estimate the median, depends on the noise parameter $\kappa$. When the knowledge of $\kappa$ is not provided, how can the algorithm be implemented?

Please also kindly refer to the Weakness section.

**Details Of Ethics Concerns:**

None. It is a theoretical paper with no ethics concerns.

---

> ### Author Response · Authors · 2024-11-23
>
> Dear Reviewer Jooe, thank you for your comments. Here are the answers to your questions and comments.
>
> **It would be great if the authors can highlight the additional technical challenges and novelties. Otherwise, this paper looks incremental given the prior works.**
>
> We addressed this question in the general response.
>
> **The motivation for the symmetric assumption is not well explained. As the symmetry in the noise is critical for the median estimate, a motivation is needed to support this assumption.**
>
>  Symmetric noise assumption may look restrictive, but we must highlight that a huge part of previous research on MAB and ZO methods with explicit noise structure is focused on symmetric distributions such as normal distribution, uniform distribution and Bernoulli distribution with p=1/2, Symmetric $\alpha$-stable distributiosn.
>
> In practice, we can not know the noise distribution for sure. Nevertheless, one can run our method and if noise is close to symmetric, our methods are to show much better convergence. Moreover, in experiments with non-symmetric noise D.2.1, our methods do not lose to the standard SOTA approaches for any noises. Hence, running our methods in practice ends up with either typical convergence rates or faster rates in symmetric cases.
>
> In the Limitation section, we describe the common strategy to solve a general optimization problem: run several algorithms in a competitive manner to see which performs better in practice. In this concept, we believe that including our methods in a list can only benefit the final result.
>
> **As this paper adopts bandits as an application and there is already some work in the bandits literature which considers heavy-tailed noise or robust bandits problems (e.g., [1]) without the symmetric noise assumption, it would be great if the authors can compare their results to them, in terms of the assumptions, methods, theoretical guarantees and efficiency of the algorithms.**
>
> We will add extra comparison dedicated to bandits methods in the revised version of the paper. The main difference is that our method can work with extremely heavy-tailed noise with unbounded math expectation ($\kappa < 1$) in both theory and practice without degeneration depending on $\kappa$. Other works do not consider that kind of noises.
>
> **Can the authors kindly modify Line 73? It can be misleading as Bubeck et al. (2013) does not make the symmetric noise assumption.**
>
> Thank you! We will follow your suggestion.
>
> **Q1:**
>
> In our proof, an extra factor $\log d$ arises since we work with the $\ell_\infty$ norm instead of $\ell_2$ norm in the proof for bandits. Lemma 1 states that we need this extra factor $a_\infty \approx \log d$ for the bound of median estimator’s variance.
>
> We would like to highlight that “optimality” here refers to the setting with **any** heavy-tailed noise with bounded variance. And we notice the fact that our bounds for any $\kappa$ and these optimal bounds exactly match. For the symmetric noises only, as far as we are aware, there is no proved lower bound.
>
> We also discussed the "suboptimality" term in the general response.
>
> **Q2:**
>
> For the most common cases $\kappa$ is at least 1 (i.e. expectation exists), hence we could take a median size $ m = 3$. In case when $\kappa$ could be arbitrarily small, we can construct an adaptive scheme, but we didn’t cover this part in our work. In practice, we additionally grid searched the median parameter $m$ among the range [3,5,7]. We noticed that unlike the choice of the clipping level, the choice of the median size slightly affects the convergence and does not require careful fine-tuning for the considered $\kappa$.  Usually, the range [3,5,7] is enough to find optimal median size for optimal convergence.

---

### Official Review · Reviewer_ZxE4 · 2024-10-31

**Soundness:** 3
**Presentation:** 3
**Contribution:** 3
**Rating:** 6
**Confidence:** 3

**Summary:**

This paper considers the problem of non-smooth convex optimization with bandit and heavy tailed feedbacks, and introduces the technique of building unbiased median gradient estimate with bounded second moment. This technique are further applied into the MAB problems with heavy-tailed rewards. Several empirical investigations are conducted to verify their theoretical findings.

**Strengths:**

- This paper is clearly written and well-organized.
- The authors have established a series of theoretical results, and perform the corresponding numerical experiments.

**Weaknesses:**

- It seems that the proposed methods are simple combinations of existing algorithms with a median gradient estimate. For example, the proposed ZO-clipped-med-SSTM is similar to ZO-clipped-SSTM, with only a different gradient estimate. While I understand that even this combination presents certain analytical challenges, I encourage the authors to clearly describe these challenges
- Currently, the effectiveness of the method is verified on synthetic datasets. Could it also be validated on real-world datasets?
- Several crucial related works on bandits with heavy-tailed feedbacks [1, 2, 3] and  non-smooth optimization [4, 5, 6] are missing from the discussion of this paper.


[1] Multi-armed bandit problems with heavy-tailed reward distributions. 2011.

[2] Almost Optimal Algorithms for Linear Stochastic Bandits with Heavy-Tailed Payoffs. 2018.

[3] Optimal Algorithms for Lipschitz Bandits with Heavy-tailed Rewards. 2019.

[4] Complexity of finding stationary points of nonconvex nonsmooth functions. 2020.

[5] No dimension-free deterministic algorithm computes approximate stationarities of Lipschitzians. 2024.

[6] High-Probability Bound for Non-Smooth Non-Convex Stochastic Optimization with Heavy Tails. 2024.

**Questions:**

See in Weakness.

---

> ### Author Response · Authors · 2024-11-23
>
> Dear Reviewer ZxE4, thank you for your comments. Here are the answers to your questions and comments.
>
> **It seems that the proposed methods are simple combinations of existing algorithms with a median gradient estimate. For example, the proposed ZO-clipped-med-SSTM is similar to ZO-clipped-SSTM, with only a different gradient estimate. While I understand that even this combination presents certain analytical challenges, I encourage the authors to clearly describe these challenges**
>
> We addressed this weakness in the general response.
>
> **Currently, the effectiveness of the method is verified on synthetic datasets. Could it also be validated on real-world datasets?**
>
>
> For the bandits' setup, we verified our method on both synthetic and real-world's cryptocurrency portfolio selection problems in Sections $5.1, 5.2$, respectively.
>
> For ZO methods, as far as we are aware, there are no publicly available real-world datasets to test performance in the community.
>
>
> **Several crucial related works on bandits with heavy-tailed feedbacks [1, 2, 3] and non-smooth optimization [4, 5, 6] are missing from the discussion of this paper.**
>
> Thank you for your comment. We will cite all mentioned works and discuss them in the revised version. Here we give brief comments on them.   We didn’t know about [1], we will enlist it. Works [2] and [3] related to contextual multi-armed bandits (linear and Lipschitz, respectively), which are out of scope of this paper.
> Works [4,5,6] focus on non-convex case, while we work with convex functions for which gradient smoothing framework has theoretical base.
>
> We believe that it is possible to expand our ZO and MAB median clipping approach to above-mentioned settings in future research.

---

### Official Review · Reviewer_uwXz · 2024-11-03

**Soundness:** 3
**Presentation:** 2
**Contribution:** 2
**Rating:** 5
**Confidence:** 4

**Summary:**

This work makes the first step to study zeroth-order non-smooth convex optimization with heavy-tailed noises. To this end, the authors propose the ZO-clipped-med-SSTM estimator and apply it to solve zeroth-order non-smooth convex optimization with heavy-tailed noises in both the constrained and unconstrained cases. Also, the authors showcase the capability of their method in the case of multi-armed bandits (MABs) with heavy-tailed reward noises. Numerical experiments are provided.

**Strengths:**

1.	This work seems to be the first work to solve the zeroth-order non-smooth convex optimization with heavy-tailed noises.

**Weaknesses:**

1. **Technical Innovations**: Both the clipping operation, the construction of the batch gradient estimator, and the algorithmic designs of ZO-clipped-med-SSTM seem particularly similar to those in [1]. Also, Lemma 1, the key lemma in this work, is proven based on the
Lemma 2.3 of [2], which guarantees the bounded second moment. I would suggest the authors give more detailed comparisons with previous works, especially on the key technical innovations that do not appear in previous works.
2. **Adversarial Case**: The proposed algorithm requires the construction of a gradient estimator in a batched manner, which makes it unable to deal with the adversarial MABs with heavy-tailed noises. This is a drawback compared with [3], which is unable to handle adversarial MABs with heavy-tailed noises.
3. **Known $\kappa$**: All the algorithms in this work seem to require the knowledge of $\kappa$ to tune the median size parameter $m$. However, the algorithm in [3] can work in the case with unknown $\kappa$.
4. **Presentation**:
   * For Table 1, I think it would be better if there are some discussions about both the advantages and the disadvantages of the results of the proposed methods against the baseline methods. Currently, the authors just present the results without providing any discussions on them.
   * Assumption 3 seems to be quite different from the heavy-tailed assumption in [3]. I think it would be better if the authors could give detailed discussions on this.

[1] Kornilov et al. Accelerated Zeroth-order Method for Non-Smooth Stochastic Convex Optimization Problem with Infinite Variance. NeurIPS, 23.

[2] Kornilov et al. Gradient-free methods for non-smooth convex stochastic optimization with heavy-tailed noise on convex compact. Computational Management Science, 23.

[3] Huang et al. Adaptive Best-of-Both-Worlds Algorithm for Heavy-Tailed Multi-Armed
Bandits. ICML, 22.

____
**Post-rebuttal**: Sorry for the late replies. I thank the authors for the explanations. Some of my previous concerns regarding the presentation are addressed. However, my concern regarding whether the proposed algorithm is applicable in adversarial cases remains unsolved. In particular, the authors said that “algorithm will make a few shots with the same distribution to be able to construct a batch”. However, in adversarial cases, I think the key problem is exactly that the loss distribution may vary in each round and I believe it is impossible to construct a batch for the loss distribution in one round. For the technical novelty and my previous concerns regarding the advantages of the results against those in [1], it turns out that these are mainly due to the symmetric noise distribution assumption in Assumption 3 of this work. Such an assumption seems not very natural in practice and is also not required by [1]. As such, I am afraid that this work might not be ready for publication in ICLR and I have to maintain my current rating for this work.

[1] Huang et al. Adaptive Best-of-Both-Worlds Algorithm for Heavy-Tailed Multi-Armed Bandits. ICML, 22.

**Questions:**

1. What does “ZO-clipped-med-SSTM” denote for?
2. On Line 14-Line 15, the authors state that “Unlike the existing high-probability results requiring the noise to have bounded $\kappa$-th moment with $\kappa\in(1, 2]$..”. However, on Line 73-Line 74, it seems that this work actually needs the assumption of bounded $\kappa$-moment of losses ($\kappa\in(1, 2]$) in the case of heavy-tailed MABs.
3. On Line 81, the authors state that the proposed method achieves high-probability optimal convergence rates. However, I do not seem to find any discussions about the lower bound of the convergence rate for the case of unconstrained optimization. How can we regard the upper bound as “optimal” if there are no lower bounds provided?
4. On Line 84-Line 85, the authors say that they achieve the $\widetilde{O}(\sqrt{dT})$ regret while also saying that their result is “suboptimal”. Why is this result “suboptimal”? To me, this is (nearly) optimal as it is known that the minimax-optimal regret lower bound for MABs is exactly of $\Omega(\sqrt{dT})$.
5. The algorithm in [3] can only obtain a regret of order $ O\left(\sigma K^{1-1 / \kappa} T^{1 / \kappa}+\sqrt{K T}\right) $. Intuitively, why the algorithm in this work can obtain the $\widetilde{O}(\sqrt{dT})$ regret without the dependence on $\kappa$ of the exponent of $T$? Is this because the algorithm explicitly leverages the knowledge of $\kappa$ to tune $m$?
6. The algorithm in [3] can only construct a biased estimator, which also results in their need to handle additional “skipping errors”. Intuitively, why the algorithm in this work can construct the unbiased estimator?

---

> ### Author Response · Authors · 2024-11-23
>
> Dear Reviewer uwXZ, thank you for your comments. Here are the answers to your questions and comments.
>
> **Technical Innovations:**
>
> We addressed this weakness in the general response.
>
> **Adversarial Case:**
>
> We do not think that batching restricts the use of our algorithm for adversarial MAB. In simple terms, instead of sampling one shot with a given probability distribution over arms, the algorithm will make a few shots with the same distribution to be able to construct a batch. This is a trade when we use more samples to construct better gradient estimates.
>
> **Known $\kappa$:**
>
> Note that we need to $\kappa$ only to set the optimal median size $m = \frac{2}{\kappa} + 1$. For the most common cases $\kappa$ is at least 1 (i.e. expectation exists), hence we could take median size m = 3. In case when $\kappa$ could be arbitrarily small, we can construct an adaptive scheme (we believe that it is possible to build analogue of scheme from [4] applied to median size), but we didn’t cover this part in our work.
>
> In experiments with bandits and ZO methods, we additionally grid searched the median parameter $m$ among the range [3,5,7]. We noticed that unlike the choice of the clipping level, the choice of the median size slightly affects the convergence and does not require careful fine-tuning for the considered $kappa$. Usually, the range [3,5,7] is enough to find optimal median size for optimal convergence.
>
> **Presentation:**
>
> 1) For Table 1, we will add discussion in the revised version of the paper. Our algorithms for ZO and bandits settings successfully incorporate the symmetry of the noise, allowing it to work with bounded $\kappa$-th moment. For $\kappa \leq1$, we are the first to prove convergence rates and verify it in practice. For convex ZO problems with $\kappa > 1$, we derive the bound $ \sim \varepsilon^{-2}$ instead of baselines $\sim \varepsilon^{-\frac{\kappa}{\kappa - 1}}$.
>
> However, such breaking results can be guaranteed only for symmetric noises. In practice, we can not know the noise distribution for sure, but one can run our method and if noise is close to symmetric, our methods are to show much better convergence. Moreover, in experiments with non-symmetric noise D.2.1, our methods do not lose to the baselines. Hence, running our methods in practice ends up with either typical convergence rates or faster rates in symmetric cases.
>
>
> 2) We discussed our Assumption in Appendix A in more detail. We show that it includes typical practical examples of heavy-tailed ZO oracles  (Remark 5) and is consistent with previous Assumptions from [1] (Remark 3). We will expand the discussion right after Assumption 3 in the revised version.
>
> **Q1:**
>
> It stands for Zero Order clipped median Stochastic Similar Triangles Method. Clipped-Stochastic Similar Triangles Method was initially presented in [2].
>
>
>
> **Q2:**
>
> We think there is a misunderstanding here.  Line 73- Line 74 refer to previous works on MAB with heavy tails. We will state it more clearly.
>
>
> **Q3:**
>
> We think there is a slight misunderstanding. For the symmetric noises, as far as we are aware, there are no proved lower bounds. On line 81, we refer to the optimal lower bounds for unconstrained zero-order optimization under **any** noise from lines 43-46 (when $\kappa = 2$). We make this moment more clear in the revised version.
>
>
> **Q4:**
>
> We addressed this question in the general response.
>
> **Q5:**
>
> The symmetry assumption makes the estimator unbiased (Lemma 1) with the finite variance. Here is the intuition behind it.  The median of $2m+1$ elements is an $m+1$-th order statistic whose distribution is the product of original heavy-tailed distributions. Hence, if $m$ is large enough, this product properly lightens the original tails.
>
> **Q6:**
>
> Intuitively, our method does not have $\kappa$ dependency due to symmetry assumption. The symmetry allows us to prove that median estimate with a few samples is unbiased estimation with the finite variance for any $\kappa > 0$. In [4], the authors use clipping to cope with heavy tails and, thus, have to leverage between low variance and bias and big number of required samples. This trade-off worsens the regret bound as $\kappa \to 1$.
>
>
> **References:**
>
> [1] Kornilov et al. Accelerated Zeroth-order Method for Non-Smooth Stochastic Convex Optimization Problem with Infinite Variance. NeurIPS, 23.
>
> [2] Gorbunov, Eduard, Marina Danilova, and Alexander Gasnikov. "Stochastic optimization with heavy-tailed noise via accelerated gradient clipping." Advances in Neural Information Processing Systems 33 (2020): 15042-15053.
>
> [3] Bubeck, S., Cesa-Bianchi, N., & Lugosi, G. (2013). Bandits with heavy tail. IEEE Transactions on Information Theory, 59(11), 7711-7717.
>
> [4] Huang et al. Adaptive Best-of-Both-Worlds Algorithm for Heavy-Tailed Multi-Armed Bandits. ICML, 22.

---

### Official Review · Reviewer_SJFH · 2024-11-04

**Soundness:** 4
**Presentation:** 3
**Contribution:** 3
**Rating:** 6
**Confidence:** 3

**Summary:**

This paper consider non-smooth convex optimization with a zeroth-order oracle with heavy-tail symmetric stochastic noise. Algorithms are proposed for settings with heavier noise than what is often considered in the literature. The technique is further applied to stochastic multi-armed bandit problem.

**Strengths:**

In the convex optimization setting, convergence rate of the proposed algorithms are either the first result or matches the best known bounds in the literature (in the case of bounded variance).

Algorithms are demonstrated to perform well in comparison to previous SOTA, even in the bandit case.

**Weaknesses:**

The regret bound in the bandit case is suboptimal.

**Questions:**

The need for a two-point oracle seems like a limiting factor. Being able to obtain 2 samples with the same noise parameter does not seem trivial. In what real world settings do we have access to such oracles?

---

> ### Author Response · Authors · 2024-11-23
>
> Dear Reviewer SJFH, thank you for your comments. Here are the answers to your questions and comments.
>
>
> **The regret bound in the bandit case is suboptimal.**
>
> We addressed this question in the general response.
>
>
> **The need for a two-point oracle seems like a limiting factor. Being able to obtain 2 samples with the same noise parameter does not seem trivial. In what real world settings do we have access to such oracles?**
>
> As an example of the real world problem for the Lipschitz oracle, we can refer to the rounding error. As two points get close to each other, the corresponding rounding error decreases. In our paper, we also consider an oracle providing two points with independent noises and call it “Independent” oracle. This oracle is more conventional as it just returns functions values corrupted by independent noise sample.

---

> > ### Comment · Reviewer_SJFH · 2024-11-27
> >
> > Thank you for the response.
> > I have 2 further comments on the presentation:
> >
> > 1. The definition of regret is stated as if actions are $x_t$ and belong to the simplex. The lines 334-341 should be rewritten and the regret definition should be formulated like a classical MAB problem. As it is now, they make it seem like the problem is a linear bandit. First the MAB problem should be completely introduced, then other notions such at $l_t$ should be discussed.
> >
> > 2. It is not meaningful for Algorithm 3 to have an output. Its output is the action it takes. The last line in the algorithm should be removed.

---

> > > ### Author Response · Authors · 2024-11-28
> > >
> > > Thank you for your comment. We fixed these presentation issues in the revised version of the paper.

---

> > > > ### Comment · Reviewer_SJFH · 2024-11-30
> > > >
> > > > Thank you for your response. I maintain my positive score.

---

### Author Response · Authors · 2024-11-28
**Revised paper content**

Dear reviewers, we have uploaded a revised version of our paper. The main edits to the paper are highlighted with the **blue color (newly added)**. These edits include:

1) **[reviewers uwXz, ZxE4, Jooe]** Expanded description of our theoretical novelty in Contributions Section 1.1 and comparison of previous and our oracles assumptions in Chapter 3.1.1.

2) **[reviewers SJFH, Jooe]** Updated introduction and related works discussion for MAB problem in Section 4.

3) **[reviewers uwXz, Jooe]** Paragraph about tuning the median size $m$ in experiments in Section 5.3

4) **[reviewers uwXz, Jooe]** Extra discussion in Limitation Section 6.1 addressing concerns about symmetrical noise in practice and necessity to know parameter $\kappa$.

5) **[reviewers uwXz, Jooe]** Explicit comparison of our methods and previous works in Section 6.2.

6) Minor updates throughout the paper for the questions we answered here, e.g., suboptimality formulation, missed papers, presentation etc.

---

### Meta-Review · Area_Chair_hTzE · 2024-12-21

**Metareview:**

This paper studies zeroth-order non-smooth convex optimization with heavy-tailed symmetric noise. Compared to previous algorithms, the algorithm proposed in this paper could handle heavier noise, i.e., noise with bounded $\kappa$-th moment for any $\kappa>0$. The authors further apply this technique to stochastic multi-armed bandit with heavy-tailed rewards to showcase the capability of their method.

The main weakness raised by the reviewers is the main assumption made in this paper, i.e., the noise symmetry assumption. Compared to prior work, the main technical innovations come from this assumption, and it is unclear if such strong assumption would hold in practice. Although the authors argued that their algorithm can be applied in practice even without this assumption, it is not justified with theoretical analysis. Moreover, some prior work in the robust bandits literature does not require such a strong assumption.

Given the high standards of ICLR and the weakness mentioned above, I would recommend rejecting this paper.

**Additional Comments On Reviewer Discussion:**

The reviewers raised concerns regarding the noise symmetry assumption, the technical innovation of the paper compared to existing work, and whether the new algorithm can be applied to adversarial multi-armed bandit. Although the authors provided responses which addressed some of those concerns, concerns regarding the noise symmetry assumption remain.

---

### Decision · Program_Chairs · 2025-01-22

Reject